# Recursively-Constrained Partially Observable Markov Decision Processes

**Qi Heng Ho**[1]  **Tyler Becker**[1]  **Benjamin Kraske**[1]  **Zakariya Laouar**[1]  **Martin S. Feather**[2]  **Federico Rossi**[2]

**Morteza Lahijanian**[1]                    **Zachary Sunberg**[1]

[1]Department of Aerospace Engineering Sciences, University of Colorado Boulder, Boulder, Colorado, USA
[2]Jet Propulsion Laboratory, California Institute of Technology, Pasadena, California, USA

## Abstract

Many sequential decision problems involve optimizing one objective function while imposing constraints on other objectives. Constrained Partially Observable Markov Decision Processes (C-POMDP) model this case with transition uncertainty and partial observability. In this work, we first show that C-POMDPs violate the optimal substructure property over successive decision steps and thus may exhibit behaviors that are undesirable for some (e.g., safety critical) applications. Additionally, online re-planning in C-POMDPs is often ineffective due to the inconsistency resulting from this violation. To address these drawbacks, we introduce the Recursively-Constrained POMDP (RC-POMDP), which imposes additional history-dependent cost constraints on the C-POMDP. We show that, unlike C-POMDPs, RC-POMDPs always have deterministic optimal policies and that optimal policies obey Bellman's principle of optimality. We also present a point-based dynamic programming algorithm for RC-POMDPs. Evaluations on benchmark problems demonstrate the efficacy of our algorithm and show that policies for RC-POMDPs produce more desirable behaviors than policies for C-POMDPs.

## 1  INTRODUCTION

Partially Observable Markov Decision Processes (POMDPs) are powerful models for sequential decision making due to their ability to account for transition uncertainty and partial observability. Their applications range from autonomous driving Pendleton et al. [2017] and robotics to geology Lauri et al. [2023], Wang et al. [2022], asset maintenance Papakonstantinou and Shinozuka [2014], and human-computer interaction Chen et al. [2020]. Constrained POMDPs (C-

POMDPs) are extensions of POMDPs that impose a bound on expected cumulative costs while seeking policies that maximize expected total reward. C-POMDPs address the need to consider multiple objectives in applications such as autonomous rover that may have a navigation task as well as an energy usage budget, or human-AI dialogue systems with constraints on the length of dialogues. However, we observe that optimal policies computed for C-POMDPs exhibit pathological behavior in some problems, which can be opposed to the C-POMDP's intended purpose.

**Example 1** (Cave Navigation). *Consider a rover agent in a cave with two tunnels, A and B, which may have rocky terrains. Traversing tunnel A has a higher expected reward than traversing tunnel B. To model wheel damage, a cost of $10$ is given for traversing through rocky terrain, and $0$ otherwise. The agent has noisy observations (correct with a probability of $0.8$) of a tunnel's terrain type, and hence, has to maintain a* belief *(probability distribution) over the terrain type in each tunnel. The task is to navigate to the end of a tunnel while ensuring that the expected total cost is below a threshold of $5$. The agent has the initial belief of $0.5$ probability of rocks and $0.5$ probability of no rocks in tunnel A, and $0$ probability of rocks and $1.0$ probability of no rocks in tunnel B.*

In this example, suppose the agent receives an observation that leads to an updated belief of $0.8$ probability that tunnel $A$ is rocky. Intuitively, the agent should avoid tunnel $A$ since the expected cost of navigating it is $8$, which violates the cost constraint of $5$. However, an optimal policy computed from a C-POMDP decides to go through the rocky region, violating the constraint and damaging the wheels. Such behavior is justified in the C-POMDP framework by declaring that, due to a low probability of observing that tunnel $A$ is rocky in the first place, the expected cost from the initial time step is still within the threshold, and so this policy is admissible. However, this pathological behavior is clearly unsuitable especially for some (e.g., safety-critical) applications.

In this paper, we first provide the key insight that the patho-

logical behavior is caused by the violation of the optimal substructure property over successive decision steps, and hence violation of the standard form of Bellman's Principle of Optimality (BPO). To mitigate the pathological behavior and preserve the optimal substructure property, we propose an extension of C-POMDPs through the addition of history-dependent cost constraints at each reachable belief, which we call Recursively-Constrained POMDPs (RC-POMDPs). We prove that deterministic policies are sufficient for optimality in RC-POMDPs and that RC-POMDPs satisfy BPO. These results suggest that RC-POMDPs are highly amenable to standard dynamic programming techniques, which is not true for C-POMDPs. RC-POMDPs provide a good balance between the BPO-violating expectation constraints of C-POMDPs and constraints on the worst-case outcome, which are overly conservative for POMDPs with inherent state uncertainty. Then, we present a point-based dynamic programming algorithm to approximately solve RC-POMDPs. Experimental evaluation shows that the pathological behavior is a prevalent phenomenon in C-POMDP policies, and that our algorithm for RC-POMDPs computes polices which obtain expected cumulative rewards competitive with C-POMDPs without exhibiting such behaviors.

In summary, this paper contributes (i) an analysis that C-POMDPs do not exhibit the optimal substructure property over successive decision steps and its consequences, (ii) the introduction of RC-POMDPs, a novel extension of C-POMDPs through the addition of history-dependent cost constraints, (iii) proofs that all RC-POMDPs have at least one deterministic optimal policy, satisfy BPO, and the Bellman operator has a unique fixed point under suitable initializations, (iv) a dynamic programming algorithm for RC-POMDPs, and (v) a series of illustrative benchmarks to demonstrate the advantages of RC-POMDPs.

**Related Work** Several solution approaches exist for C-POMDPs with expectation constraints de Nijs et al. [2021]. These include offline Isom et al. [2008], Kim et al. [2011], Poupart et al. [2015], Walraven and Spaan [2018], Kalagarla et al. [2022], Wray and Czuprynski [2022] and online methods Lee et al. [2018], Jamgochian et al. [2023]. These works suffer from the unintuitive behavior discussed above. This paper shows that this behavior is rooted in the violation of optimal substructure by C-POMDPs and proposes a new problem formulation that obeys BPO.

BPO violation has also been discussed in fully-observable Constrained MDPs (C-MDPs) with state-action frequency and long-run average cost constraints Haviv [1996], Chong et al. [2012]. To overcome it, Haviv [1996] proposes an MDP formulation with sample path constraints. In C-POMDPs with expected cumulative costs, this BPO-violation problem remains unexplored. Additionally, adoption of the MDP solution of worst-case sample path constraints would be overly conservative for POMDPs, which

are inherently characterized by state uncertainty. This paper fills that gap by studying the BPO of C-POMDPs and addressing it by imposing recursive expected cost constraints.

From the algorithmic perspective, the closest work to ours is the C-POMDP point-based value iteration (CPBVI) algorithm Kim et al. [2011]. Samples of admissible costs, defined by Piunovskiy and Mao [2000] for C-MDPs, are used with belief points as a heuristic to improve computational tractability of point-based value iteration for C-POMDPs. However, since CPBVI is designed for C-POMDPs, the synthesized policies by CPBVI may still exhibit pathological behavior. In this paper, we formalize the use of history-dependent expected cost constraints and provide a thorough analysis of it. We show that this problem formulation eliminates the pathological behavior of C-POMDPs.

## 2 CONSTRAINED POMDPS

POMDPs model sequential decision making problems under transition uncertainty and partial observability.

**Definition 1** (POMDP). *A Partially Observable Markov Decision Process (POMDP) is a tuple $\mathcal{P} = (S, A, O, T, R, Z, \gamma, b_0)$, where: $S, A$, and $O$ are finite sets of states, actions and observations, respectively, $T : S \times A \times S \rightarrow [0, 1]$ is the transition probability function, $R : S \times A \rightarrow [R_{min}, R_{max}]$, for $R_{min}, R_{max} \in \mathbb{R}$, is the immediate reward function, $Z : S \times A \times O \rightarrow [0, 1]$ is the probabilistic observation function, $\gamma \in [0, 1)$ is the discount factor, and $b_0 \in \Delta(S)$ is an initial belief, where $\Delta(S)$ is the probability simplex (the set of all probability distributions) over $S$.*

We denote the probability distribution over states in $S$ at time $t$ by $b_t \in \Delta(S)$ and the probability of being in state $s$ at time $t$ by $b_t(s)$.

The evolution of an agent according to a POMDP model is as follows. At each $t \in \mathbb{N}_0$, the agent has a belief $b_t$ of its state $s_t$ as a probability distribution over $S$ and takes action $a_t \in A$. Its state evolves from $s_t \in S$ to $s_{t+1} \in S$ according to $T(s_t, a_t, s_{t+1})$, and it receives an immediate reward $R(s_t, a_t)$ and observation $o_t \in O$ according to observation probability $Z(s_{t+1}, a_t, o_t)$. The agent then updates its belief using Bayes theorem; that is for $s_{t+1} = s'$,

$$b_{t+1}(s') \propto Z(s', a_t, o_t) \sum_{s \in S} T(s, a_t, s') b_t(s). \quad (1)$$

Then, the process repeats. Let $h_t = \{a_0, o_0, \cdots, a_{t-1}, o_{t-1}\}$ denote the history of the actions and observations up to but not including time step $t$; thus, $h_0 = \emptyset$. The belief at time step $t$ is therefore $b_t = P(s_t \mid b_0, h_t)$. For readability, we do not explicitly include $b_0$, as all variables are conditioned on $b_0$.

The agent chooses actions according to a policy $\pi : \Delta(S) \to \Delta(A)$, which maps a belief $b$ to a probability distribution over actions. $\pi$ is called *deterministic* if $\pi(b)$ is a unitary distribution for every $b \in \Delta(S)$. A policy is typically evaluated according to the expected rewards it accumulates over time. Let $R(b,a) = \mathbb{E}_{s \sim b}[R(s,a)]$ be the expected reward for the belief-action pair $(b,a)$. The *expected discounted sum of rewards* that the agent receives under policy $\pi$ starting from belief $b_t$ is

$$V_R^\pi(b_t) = \mathbb{E}_{\pi,T,Z}\Big[ \sum_{\tau=t}^\infty \gamma^{\tau-t} R\left(b_\tau, \pi(b_\tau)\right) \mid b_t \Big]. \quad (2)$$

Additionally, the $Q$ reward-value is defined as

$$Q_R^\pi(b_t, a) = R(b_t, a) + \gamma \, \mathbb{E}_{T,Z}[V_R^\pi(b_{t+1})]. \quad (3)$$

The objective of POMDP problems is often to find a policy that maximizes $V_R^\pi(b_0)$.

As an extension of POMDPs, Constrained POMDPs add a constraint on the expected cumulative costs.

**Definition 2** (C-POMDP). *A* Constrained POMDP (C-POMDP) *is a tuple $\mathcal{M} = (\mathcal{P}, C, \hat{c})$, where $\mathcal{P}$ is a POMDP as in Def. 1, $C : S \times A \to \mathbb{R}_{\geq 0}^n$ is a cost function that maps each state action pair to an $n$-dimensional vector of non-negative costs, and $\hat{c} \in \mathbb{R}_{\geq 0}^n$ is an $n$-dimensional vector of expected cost thresholds from the initial belief state $b_0$.*

In C-POMDPs, by executing action $a \in A$ at state $s \in S$, the agent receives a cost vector $C(s,a)$ in addition to the reward $R(s,a)$. Let $C(b,a) = \mathbb{E}_{s \sim b}[C(s,a)]$. The expected sum of costs incurred by the agent under $\pi$ from belief $b_t$ is:

$$V_C^\pi(b_t) = \mathbb{E}_{\pi,T,Z}\Big[ \sum_{\tau=t}^\infty \gamma^{\tau-t} C\left(b_\tau, \pi(b_\tau)\right) \mid b_t \Big]. \quad (4)$$

Additionally, the $Q$ cost-value is defined as

$$Q_C^\pi(b_t, a) = C(b_t, a) + \gamma \, \mathbb{E}_{T,Z}[V_C^\pi(b_{t+1})]. \quad (5)$$

In C-POMDPs, the constraint $V_C^\pi(b_0) \leq \hat{c}$, where $\leq$ refers to the component-wise inequality, is imposed on the POMDP optimization problem as formalized below.

**Problem 1** (C-POMDP Planning Problem). *Given a C-POMDP, compute policy $\pi^*$ that maximizes total expected reward in Eq. (2) from initial belief $b_0$ while the total expected cost vector in Eq. (4) is bounded by $\hat{c}$, i.e.,*

$$\pi^* = \arg\max_\pi V_R^\pi(b_0) \quad s.t. \quad V_C^\pi(b_0) \leq \hat{c}. \quad (6)$$

Unlike POMDPs that have at least one deterministic optimal policy Sondik [1978], optimal policies of C-POMDPs may require randomization, and hence there may not exist an optimal deterministic policy Kim et al. [2011].

Next, we discuss why the solutions to Problem 1 may not be desirable and an alternate formulation is necessary.

## 2.1 OPTIMAL SUBSTRUCTURE PROPERTY

A problem has the optimal substructure property if *an optimal solution to the problem contains optimal solutions to its subproblems* Cormen et al. [2009]. Additionally, Cormen et al. note that these subproblems must be independent of each other. If this holds for Problem 1, then the optimal policy $\pi^*(b_0)$ at $b_0$ can be computed recursively by finding the optimal policy $\pi^*(h_t)$ for each successive history *for the same planning problem*. Thus, a natural subproblem to Eq. (6) is the history-based subproblem $(\mathcal{M}, h_t)$, with $\pi^*(h_t) = \arg\max_\pi V_R^\pi(h_t)$ s.t. $V_C^\pi(b_0) \leq \hat{c}$[1]. We show that this subproblem violates the optimal substructure property, which makes the employment of standard dynamic programming techniques difficult[2].

Since the constraint of Eq. (6) is defined only at $b_0$, the subproblem at $h_t$ must consider the expected cumulative cost of the policy from $b_0$. It is not enough to compute the expected total cost obtained from $b_0$ to $h_t$, as an optimal cost-value from $h_t$ depends on cost-values of other subproblems. We illustrate this with an example. Consider the POMDP (depicted as a belief MDP) in Figure 1, which is a simplified version of Example 1. W.l.o.g., let $\gamma = 1$. The agent starts at $b_0$ with constraint $\hat{c} = 5$. Actions $a_A$ and $a_B$ represent going through tunnels A and B, and $r$ and $nr$ are the observations that tunnel A is rocky and not rocky, respectively.

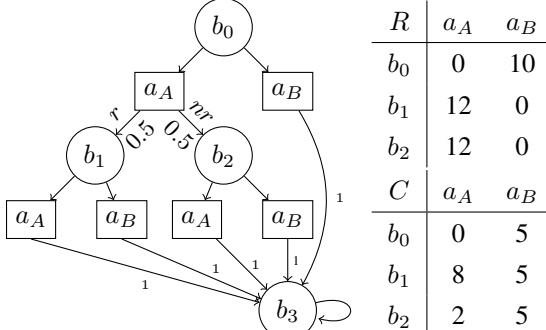

Figure 1: Counter-example POMDP with associated reward and cost functions. The action at $b_3$ has 0 reward and cost.

By examining the reward function, we see that action $a_A$ returns the highest reward everywhere except $b_0$. Action $a_B$ returns a higher reward at $b_0$. Let $\pi_A$ be the policy that chooses $a_A$ at every belief, and $\pi_B$ the one that chooses $a_B$ at $b_0$. The cost-values for these policies are $V_C^{\pi_A}(b_0) = V_C^{\pi_B}(b_0) = 5 \leq \hat{c}$, and the reward-values are $V_R^{\pi_A}(b_0) = 12, V_R^{\pi_B}(b_0) = 10$. Note that both policies satisfy the constraint and any policy that chooses $a_B$ at $b_1$ or $b_2$, or that randomizes between $\pi_A$ and $\pi_B$ has value less than

---

[1]Constraining $V_C^\pi(h_t) \leq \hat{c}$ also violates the property as the constraint is defined only at $b_0$.

[2]Some approaches use dynamic programming (Isom et al. [2008], Kim et al. [2011]), but they do not find optimal policies.

$V_R^{\pi_A}(b_0)$; hence, $\pi_A$ is the optimal policy. However, when planning at $b_1$, i.e., $h_1$, it is impossible to decide that $a_A$ is optimal without first knowing that action $a_A$ at $h_2$ incurs 2 cost and is optimal. The decisions at $b_1$ and $b_2$ cannot be computed separately as subproblems.

To get around this dependence, we can include information about how much cost the policy incurs at other subproblems and how much cost policies can incur from $h_t$, obtaining a *policy-dependent* subproblem $(\mathcal{M}, h_t, \pi)$. This subproblem definition exhibits the optimal substructure property only if we relax the restriction of subproblems being independent. Nonetheless, the optimal solution to a subproblem $(\mathcal{M}, h_t, \pi)$ is only guaranteed to be optimal for the full problem if an optimal policy $\pi^*$ is already provided.

### 2.1.1 Pathological Behavior : Stochastic Self-Destruction

A main consequence of history-dependent subproblems violating the optimal substructure property and instead requiring policy-dependent subproblems is that optimal policies may exhibit unintuitive behaviors during execution.

In the above example, the optimal policy from $b_0$ first chooses action $a_A$. Suppose that $h_1$ is reached. The cost constraint at $b_1$ remains at 5 since no cost has been incurred. However, the optimal C-POMDP policy chooses action $a_A$ and incurs a cost of 8 which violates the constraint, even though there is another action, $a_B$, that incurs a lower expected cost that satisfies the constraint. Therefore, in 50% of executions, when $h_1$ is reached, the agent intentionally violates the cost constraint to get higher expected rewards, even if a policy that satisfies the cost constraint exists. We term this pathological behavior *stochastic self-destruction*.

This unintuitive behavior is mathematically correct in the C-POMDP framework because the policy still satisfies the constraint at the initial belief state on expectation. An optimal C-POMDP policy exploits the nature of the constraint in Eq. (6) to intentionally violate the cost constraint for some belief trajectories. A concrete manifestation of this phenomenon is in the stochasticity of the optimal policies for C-POMDPs. These policies randomize between deterministic policies that violate the expected cost threshold but obtain higher expected reward, and those that satisfy the cost threshold but obtain lower expected reward.

Another consequence is a mismatch between optimal policies planned from a current time step and optimal policies planned at future time steps. In the example in Figure 1, if re-planning is conducted at $b_1$, the re-planned optimal policy selects $a_B$ instead of $a_A$. In fact, the policy that initially takes $a_B$ at $b_0$ achieves a higher expected reward than the original policy that takes $a_A$ at $b_0$ and re-plans at future time steps. This phenomenon can therefore lead to poor performance of the closed-loop system during execution.

**Remark 1.** *We remark that the pathological behavior arises due to the C-POMDP problem formulation, and not the algorithms designed to solve C-POMDPs. Further, this issue cannot be addressed by simply restricting solutions to deterministic policies since they also exhibit the pathological behavior, as seen in the example in Figure 1.*

## 3 RECURSIVELY-CONSTRAINED POMDPS

To mitigate the pathological behaviors and obtain a (policy-independent) optimal substructure property, we aim to align optimal policies computed at a current belief with optimal policies computed at future (successor) beliefs. We propose a new problem formulation called Recursively-Constrained POMDP (RC-POMDP), which imposes additional recursively defined constraints on a policy.

An RC-POMDP has the same tuple as a C-POMDP, but with recursive constraints on beliefs at future time steps. These constraints enforce that a policy must satisfy a history dependent cumulative expected cost constraint at every future belief state. Intuitively, we bound the cost value at every belief such that the constraint in the initial node is respected.

The expected cumulative cost of the trajectories associated with history $h_t$ is given as:

$$W(h_t) = \sum_{\tau=0}^{t-1} \gamma^\tau \mathbb{E}_{s_\tau \sim b_\tau} \left[ C(s_\tau, a_\tau) \mid h_\tau \right]. \qquad (7)$$

We can direct the optimal policy at each time step $t$ by imposing that the total expected cumulative cost satisfies the initial cost constraint $\hat{c}$. For a given $h_t$ and its corresponding $b_t$, the expected cumulative cost at $b_0$ is given by:

$$V_{C|h_t}^\pi(b_0) = W(h_t) + \gamma^t V_C^\pi(b_t). \qquad (8)$$

Therefore, the following constraint should be satisfied by a policy $\pi$ at each future belief state:

$$W(h_t) + \gamma^t V_C^\pi(b_t) \le \hat{c}. \qquad (9)$$

We define the admissibility of a policy $\pi$ accordingly.

**Definition 3** (Admissible Policy). *A policy $\pi$ is k-admissible for a $k \in \mathbb{N}_0 \cup \{\infty\}$ if $\pi$ satisfies Eq. (9) for all $t \in \{0, \ldots, k-1\}$ and all histories $h_t$ of length $t$ induced by $\pi$ from $b_0$. A policy is called* admissible *if it is $\infty$-admissible.*

Since RC-POMDP policies are constrained based on history, it is not sufficient to directly use belief-based policies. Thus, we consider history-based policies in this work. A history-based policy maps a history $h_t$ to a probability over actions $\Delta(A)$.

The RC-POMDP optimization problem is formalized below.

**Problem 2** (RC-POMDP Planning Problem). *Given a C-POMDP and an admissibility constraint $k \in \mathbb{N} \cup \{\infty\}$, compute optimal policy $\pi^*$ that is $k$-admissible, i.e., $\forall h_t$,*

$$\pi^*(h_t) = \arg\max_\pi V_R^\pi(h_t) \tag{10}$$

$$s.t. \ \ W(h_t) + \gamma^t V_C^\pi(b_t) \le \hat{c} \ \ \forall t \in \{0, \ldots, k-1\}. \tag{11}$$

Note that Problem 2 is an infinite-horizon problem since the optimization objective (10) is infinite horizon. The admissibility constraint $k$ is a user-defined parameter. In this work, we focus on $k = \infty$, i.e., admissible policies.

**Remark 2.** *In POMDPs, reasoning about cost is done on expectation due to state uncertainty. C-POMDPs bound the expected total cost of state trajectories, enabling belief trajectories with low expected costs to compensate for those with high expected costs. Conversely, a worst-case constraint formulation of the problem, which never allows any violations during execution, may be overly conservative. RC-POMDPs strike a balance between the two; it bounds the expected total cost for all belief trajectories, only allowing cost violations during execution due to state uncertainty.*

## 4 THEORETICAL ANALYSIS OF RC-POMDPS

We first transform Eq. (11) into an equivalent recursive form that is better suited for policy computation, e.g., tree search and dynamic programming. By rearranging Eq. (11), $V_C^\pi(b_t) \le \gamma^{-t} \cdot (\hat{c} - W(h_t))$. Based on this, we define the *history-dependent admissible cost bound* as:

$$d(h_t) = \gamma^{-t} \cdot (\hat{c} - W(h_t)), \tag{12}$$

which can be computed recursively:

$$d(h_0) = \hat{c}, \quad d(h_{t+1}) = \gamma^{-1} \cdot \big(d(h_t) - C(b_t, a_t)\big). \tag{13}$$

Then, Problem 2 can be reformulated with recursive bounds.

**Proposition 1.** *Problem 2 can be rewritten as:*

$$\pi^* = \arg\max_\pi V_R^\pi(b_0)$$
$$s.t. \quad V_C^\pi(b_t) \le d(h_t) \quad \forall t \in \{0, 1, \ldots, k-1\}, \tag{14}$$

*where $d(h_t)$ is defined recursively in Eq. (13).*

**Optimality of Deterministic Policies** Here, we show that deterministic policies suffice for optimality in RC-POMDPs.

**Theorem 1.** *An RC-POMDP with admissibility constraint $k = \infty$ has at least one deterministic optimal policy if an admissible policy exists.*

A proof is provided in the Appendix. The main intuition is that we can always construct an optimal deterministic policy

from an optimal stochastic policy. That is, at every history in which the policy has stochasticity, we can construct a new admissible policy that achieves the same reward-value while remaining admissible by deterministically choosing one of the stochastic actions at that history. We obtain a deterministic optimal policy by inductively performing this determinization at all reachable histories.

**Satisfaction of Bellman's Principle of Optimality** Here, we show that RC-POMDPs satisfy BPO with a policy-independent optimal substructure.

**Proposition 2** (Belief-Admissible Cost Formulation). *An RC-POMDP belief $b_t$ with history dependent admissible cost bound $d(h_t)$ can be rewritten as an augmented belief-admissible cost state $\bar{b}_t = (b_t, d(h_t))$. Further, the augmented Q-values for a policy can be written as:*

$$Q_R^\pi((b_t, d(h_t)), a) = R(b_t, a) + \gamma \, \mathbb{E}[V_R^\pi((b_{t+1}, d(h_{t+1})))],$$
$$Q_C^\pi((b_t, d(h_t)), a) = C(b_t, a) + \gamma \, \mathbb{E}[V_C^\pi((b_{t+1}, d(h_{t+1})))].$$

We first see that the evolution of $\bar{b}_t$ is Markovian, i.e.,

$$P(\bar{b}_{t+1} \mid \bar{b}_t, a_t, o_t, h_t)$$
$$= \begin{cases} P(b_{t+1} \mid b_t, a, o_t, h_t) & \text{if } d(h_{t+1}) = \frac{(d(h_t) - C(b_t, a_t))}{\gamma} \\ 0 & \text{otherwise,} \end{cases}$$

thus, $P(\bar{b}_{t+1} \mid \bar{b}_t, a_t, o_t, h_t) = P(\bar{b}_{t+1} \mid \bar{b}_t, a_t, o_t)$.

Here, we use the policy iteration version of Bellman equation, but a similar argument can be made for value iteration.

**Theorem 2.** *Fix $\pi$. Let $V^\pi = (V_R^\pi, V_C^\pi)$ be reward- and cost-value function for $\pi$. The Bellman operator $\mathbb{B}$ for policy $\pi$ for an RC-POMDP is given by, $\forall \bar{b}_t$,*

$$\mathbf{a} = \arg\max_{a \in A} \Big[ Q_R^\pi(\bar{b}_t, a) \mid Q_C^\pi(\bar{b}_t, a) \le d(h_t) \Big] \tag{15}$$

$$\mathbb{B}[V^\pi](\bar{b}_t) \triangleq \begin{cases} \big(Q_R^\pi(\bar{b}_t, a), Q_C^\pi(\bar{b}_t, a)\big), \ a \in \mathbf{a} & \text{if } \mathbf{a} \ne \emptyset, \\ \big(V_R^\pi(\bar{b}_t), (\infty, \ldots, \infty)\big) & \text{if } \mathbf{a} = \emptyset, \end{cases}$$

*Assume an admissible policy exists for the RC-POMDP with admissibility constraint $k = \infty$. Let $V^{\pi^*} = (V_R^{\pi^*}, V_C^{\pi^*})$ be the values for an optimal admissible policy $\pi^*$, and we obtain a new policy $\pi'$ with $(V_R^{\pi'}, V_C^{\pi'}) = \mathbb{B}[V^{\pi^*}]$. $\pi^*$ satisfies the BPO criterion of an admissible optimal policy:*

$$V_R^{\pi'}(\bar{b}_t) = V_R^{\pi^*}(\bar{b}_t) \qquad \forall \bar{b}_t, \tag{16}$$

$$V_C^{\pi'}(b_t) \le d(h_t) \qquad \forall \bar{b}_t \in \text{REACH}^{\pi'}(\bar{b}_0), \tag{17}$$

*where $\text{REACH}^\pi(\bar{b}_0)$ is the set of augmented belief states reachable from $b_0$ under policy $\pi$.*

This theorem shows that an optimal policy remains admissible and optimal w.r.t rewards after applying $\mathbb{B}$ on a policy independent value function $V$. Note that $V_C^*$ is not unique

as there may be multiple optimal cost-value functions for an optimal $V_R^*$. Next, we show that $\mathbb{B}$ is a contraction over reward-values for a suitably initialized value function, which is one that defines the space of admissible policies.

**Theorem 3.** *For each $\bar{b}_t$, define $\Phi(\bar{b}_t)$ as the set of admissible policies from $\bar{b}_t$:*

$$\Phi(\bar{b}_t) = \{\pi \mid V_C^\pi(b_\tau) \le d(h_\tau) \ \forall \tau \ge t\}. \tag{18}$$

*$V^{\pi^0}$ is a well behaved initial value function if the following holds for all $\bar{b}_t$. If $\Phi(\bar{b}_t) = \emptyset$, $V_C^{\pi^0}(\bar{b}_t) = (\infty, \dots, \infty)$. If $\Phi(\bar{b}_t) \ne \emptyset$, $V_C^{\pi^0}(\bar{b}_t) \le d(h_t)$.*

*Suppose that $V^{\pi^0}$ is well behaved, then $\mathbb{B}^n[(V_R^{\pi^0}, V_C^{\pi^0})]_x \rightarrow (V_R^{\pi^*}, V_C^{\pi^n})$ as $n \rightarrow \infty$. That is, starting from $\pi^0$, $\mathbb{B}$ is a contraction on $V_R$ and $V_R^{\pi^*}$ is a unique fixed point.*

Proofs of all results are provided in the Appendix. Theorems 1-3 show that it is sufficient to search in the space of deterministic policies for an optimal one, and the policy-independent optimal substructure of RC-POMDPs can be exploited to employ dynamic programming for an effective and computationally efficient algorithm for RC-POMDPs. Further, Theorem 3 shows that determining policy admissibility is essential for effective dynamic programming. These results also indicate that optimal policies for RC-POMDPs do not exhibit the same pathological behaviors as C-POMDPs.

# 5 DYNAMIC PROGRAMMING FOR RC-POMDPS

With the theoretical foundation above, we devise a first attempt at an algorithm that approximately solves Problem 2 with scalar cost and admissibility constraint $k = \infty$. We leave the multi-dimensional and finite $k$ cases for future work. The algorithm is called Admissibility Recursively Constrained Search (ARCS). ARCS takes advantage of the Markovian property of the belief-admissible cost formulation in Proposition 2, and Theorems 1-3 to utilize point-based dynamic programming in the space of deterministic and admissible policies, building on unconstrained POMDP methods [Shani et al., 2013].

ARCS is outlined in Algorithm 1. It takes as input the RC-POMDP $\mathcal{M}$ and $\epsilon > 0$, a target error between the computed policy and an optimal policy at $b_0$. ARCS explores the search space by incrementally sampling points in the history space. These points form nodes in a policy tree $T$. At each iteration, a SAMPLE step expands a sequence of points starting from the root. Then, a Bellman BACKUP step is performed for each sampled node. Finally, a PRUNE step removes sub-optimal nodes. These three steps are repeated until an admissible $\epsilon$-optimal policy is found. Pseudocode

---

**Algorithm 1: Anytime Recursively Constrained Search**

ARCS ($\mathcal{M}, \epsilon$)
1: Initialize cost-minimizing policy $\hat{\pi}_c^{min} = \Gamma_c^{min}$.
2: $(\alpha_r, \alpha_c) \leftarrow \arg\min_{(\alpha_r, \alpha_c) \in \Gamma_{c_{min}}} \alpha_r^T b_0$
3: $\underline{V}_R \leftarrow \alpha_r^T b_0, \overline{V}_C \leftarrow \alpha_c^T b_0$.
4: Initialize $\overline{V}_R$ and $\underline{V}_C$ for $b_0$ with FIB.
5: Initialize $k_0$ with Eq. (19)-(20).
6: $T \leftarrow v_0 = (b_0, \hat{c}, k_0, \overline{V}_R, \underline{V}_R, \overline{V}_C, \underline{V}_C, \emptyset, \emptyset, \emptyset, \emptyset)$.
7: **repeat**
8: $\quad B_{sam} \leftarrow$ SAMPLE$(\epsilon)$.
9: $\quad$ **for all** $v \in B_{sam}$ **do**
10: $\quad\quad$ BACKUP$(v)$.
11: $\quad$ **end for**
12: $\quad$ PRUNE().
13: **until** termination conditions are satisfied
14: **return** $T, \Gamma_{c_{min}}$.

---

for SAMPLE, BACKUP and PRUNE are provided in the appendix.

**Policy Tree Representation** We represent the policy with a policy tree $T$. A node in $T$ is a tuple $v = (b, d, k, \overline{V}_R, \underline{V}_R, \overline{V}_C, \underline{V}_C, \overline{Q}_R, \underline{Q}_R, \overline{Q}_C, \underline{Q}_C)$, where $b$ is a belief, $d$ is a history-dependent admissible cost bound, $k$ is a lower bound on admissible horizon, $\overline{V}_R$ and $\underline{V}_R$ are the two-sided bounds on reward-values, $\overline{V}_C$ and $\underline{V}_C$ are the two-sided cost-value bounds, $\overline{Q}_R, \underline{Q}_R$ represent two-sided bound on $Q$ reward-value, and $\underline{Q}_C, \overline{Q}_C$ represent the two-sided bounds on $Q$ cost-value. The root of $T$ is the node $v_0$ with $b = b_0$, $d = \hat{c}$, and admissible horizon lower bound $k_0$.

From Theorem 3, a key aspect of effective dynamic programming for RC-POMDPs is computing admissible policies. This can be approximated by minimizing $V_C(\hat{b}_t)$. As a pre-processing step, we first approximate a minimum cost-value policy $\pi_c^{min} = \arg\inf_\pi V_C^\pi$. An arbitrarily tight under-approximation (upper bound) $\hat{\pi}_c^{min}$ as a set of $|S|$-dimensional hyperplanes, called $\alpha$-vectors, can be computed efficiently with a POMDP algorithm Hauskrecht [2000]. The reward-values obtained by $\hat{\pi}_c^{min}$ is also a lower bound on the optimal reward-value. Thus, $\hat{\pi}_c^{min}$ is represented by a set of $\alpha$-vector pairs $(\alpha_r, \alpha_c) \in \Gamma_C^{min}$. $\hat{\pi}_c^{min}$ is used to initialize our policy, and is used from leaf nodes of $T$.

To initialize a new node, belief $b'$ is computed with Eq. (1), and $d'$ is computed recursively with Eq. (13). We initialize $\underline{V}_R$ and $\overline{V}_C$ with $\hat{\pi}_c^{min}$, and initialize $\underline{V}_C$ and $\overline{V}_R$ independently using the Fast Informed Bound (FIB) Hauskrecht [2000], and $k'$ is a lower bound of the admissible horizon.

**Admissible Horizon Lower Bound** It is computationally intractable to exhaustively search the possibly infinite policy space. Thus, we maintain a lower bound on the admissible horizon of the policy for every node. It is used to compute admissibility beyond the current search depth of the tree, and to improve search efficiency via pruning. To initialize the

admissible horizon guarantee of a leaf node, we compute a lower bound on the admissible horizon $k$ when using $\hat{\pi}_{c_{min}}$.

**Lemma 1.** *Let the maximum 1-step cost $C_{max}$ that $\hat{\pi}_c^{min}$ incurs at each time step across the entire belief space be $C_{max} = \max_{b \in B} C(b, \hat{\pi}_c^{min}(b))$. Then, for a node $v$, if $v.d < 0$, then $k = 0$. For a leaf node $v$ with $v.d \geq 0$, $\hat{\pi}_c^{min}$ is at least $k$-admissible with*

$$k = \lfloor \log\left(1 - (v.d/C_{max}) \cdot (1 - \gamma)\right)/\log(\gamma) \rfloor, \quad (19)$$

*and $\hat{\pi}_c^{min}$ is admissible from history $h$ if*

$$C_{max}/(1 - \gamma) \leq v.d \quad or \quad v.\overline{V}_C^{\hat{\pi}_c^{min}} = 0 \leq v.d. \quad (20)$$

A proof is provided in the appendix. This lemma provides sufficient conditions for admissibility of computed policies. We compute an upper bound on the parameter $C_{max}$,

$$C_{max} \leq V_{C,max}^{\hat{\pi}_c^{min}} = \max_{b \in \Delta(S)} \min_{(\alpha_r, \alpha_c) \in \Gamma_{c_{min}}} \alpha_c^T b, \quad (21)$$

where $\alpha_c$ refers to a cost $\alpha$-vector. This can be solved efficiently with the maximin LP Williams [1990]:

$$\max_{z,b} z \quad \text{s.t.} \quad \alpha_c^T b \geq z, \quad (\alpha_r, \alpha_c) \in \Gamma, \quad b \in \Delta(S). \quad (22)$$

**Sampling**  ARCS uses a mixture of random sampling Spaan and Vlassis [2005] and heuristic search (SARSOP) Kurniawati et al. [2008]. Our empirical evaluations suggest that this approach is an effective balance between finding policies with high cumulative reward and that are admissible. At each SAMPLE step, ARCS expands the search space from the root of $T$, with either heuristic sampling or random sampling. For heuristic sampling, we use the same sampling strategy and sampling termination condition as SARSOP. It works by choosing actions with the highest $\overline{Q}_R$, and observations that have the largest contribution to the gap at the root of $T$, and sampling terminates based on a combination of selective deep sampling and a gap termination criterion. With random sampling, actions and observations are chosen randomly while traversing the tree until a new node is reached and added to the tree. Sampled points are chosen for BACKUP. **Backup**  The BACKUP operation at node $v$ updates the values in the node by back-propagating the information of the children of $v$ back to $v$. First, the values of $\overline{Q}_R, \underline{Q}_R$ and $\overline{Q}_C$ are computed for each action using Eq. (3) and Eq. (5) for rewards and costs, respectively. Then, an RC-POMDP backup Eq. (15) is used to update $\overline{V}_R, \underline{V}_R, \overline{V}_C, \underline{V}_C$. The action selected to update $\underline{V}_R$ is used to update $k$ by back-propagating the minimum $k$ of all children. If no actions are feasible, all current policies from that node are inadmissible, and we update the reward- and cost-values using the action with the minimum $Q$-cost value, and set $k = 0$.

**Pruning**

To keep the size of $T$ small and improve tractability, we prune nodes and node-actions that are suboptimal, using the following criteria. First, for each node $v \in B_{sam}$, if $v.\underline{V}_C > v.d$, no admissible policies exist from $v$, so $v$ and its subtree are pruned. Next, we prune actions as follows. Let $k(v, a)$ be the admissible horizon guarantees of the successor nodes from taking action $a$ at node $v$. Between two actions $a$ and $a'$, if $k(v, a') = \infty$ and $v.\overline{Q}_R(a) < v.\underline{Q}_R(a')$, we prune the node-action $(v, a)$ (disallow taking action $a$ at node $v$), since action $a$ can never be taken by the optimal admissible policy. Next, if all node-actions $(v, a)$ are pruned, $v$ is also pruned. Finally, the node-action $(v, a)$ is pruned if any successor node from taking $a$ at $v$ is pruned. Nodes and node-actions that are pruned are not chosen during action and observation selection during SAMPLE and BACKUP.

**Proposition 3.** *PRUNE only removes sub-optimal policies.*

**Termination Condition**  ARCS terminates when two conditions are met, (i) when it finds an admissible policy, i.e., $v_0.k = \infty$, which is when all leaf nodes $v_{leaf}$ reachable under a policy satisfy Eq. (20), and (ii) when it finds an $\epsilon$-optimal policy, i.e., when the gap criterion at the root is satisfied, that is when $v_0.\overline{V}_R - v_0.\underline{V}_R \leq \epsilon$.

**Remark 3.** *ARCS can be modified to work in an anytime fashion given a time limit, and output the best computed policy and its admissible horizon guarantee $v_0.k$.*

### 5.1   ALGORITHM ANALYSIS

Here, we analyze the theoretical properties of ARCS.

**Lemma 2** (Bound Validity)**.** *Given an RC-POMDP with admissibility constraint $k = \infty$, let $T$ be the policy tree after some iterations of ARCS. Let $V_R^*$ be the reward-value of an optimal admissible policy. At every node $v$ with $\bar{b} = (b, d)$ and admissible horizon guarantee $v.k = \infty$, it holds that: $v.\underline{V}_R \leq V_R^*(\bar{b}) \leq v.\overline{V}_R \quad and \quad v.\overline{V}_C \leq d$.*

**Theorem 4** (Soundness)**.** *Given an RC-POMDP with admissibility constraint $k = \infty$ and $\epsilon$, if ARCS terminates with a solution, the policy is admissible and $\epsilon$-optimal.*

ARCS is not complete. It may not terminate, due to conservative computation of admissible horizon and needing to search infinitely deep to find admissible policies for some problems. However, ARCS can find $\epsilon$-optimal admissible policies for many problems, such as the ones in our evaluation. These are problems where a finite depth is sufficient to compute admissibility even with conservatism. We leave the analysis of such classes of RC-POMDPs to future work.

## 6   EXPERIMENTAL EVALUATION

To the best of our knowledge, this work is the first to propose and solve RC-POMDPs, and thus there are no existing

algorithms to compare to directly. The purpose of our evaluation is to (i) empirically compare the *behavior* of policies computed for RC-POMDPs with those computed for C-POMDPs, and (ii) evaluate the performance of our proposed algorithm for RC-POMDPs. To this end, we consider the following offline algorithms to compare against our **ARCS**[3]:

- **CGCP** Walraven and Spaan [2018]: Algorithm that computes near-optimal policies for C-POMDPs using a primal-dual approach.
- **CGCP-CL**: Closed-loop CGCP with updates on belief and admissible cost at each time step.
- **Exp-Gradient Kalagarla et al. [2022]**: Algorithm that computes mixed policies using a no-regret learning approach with a primal-dual approach using an Exponentiated Gradient method.
- **CPBVI** Kim et al. [2011]: Approximate dynamic programming that uses admissible cost as a heuristic.
- **CPBVI-D**: We modify CPBVI to compute deterministic policies to evaluate its efficacy for RC-POMDPs.

Since the purpose of our comparison between RC-POMDPs and C-POMDPs is mainly with regard to constraints, we do not compare to online C-POMDP algorithms such as CC-POMCP Lee et al. [2018] which can handle larger problems but do not have anytime guarantees on constraint satisfaction.

We consider the following environments: (i) **CE**: Counterexample in Figure 1, (ii) **C-Tiger**: A Constrained version of Tiger POMDP Kaelbling et al. [1998], (iii) **CRS**: Constrained RockSample Lee et al. [2018], and (iv) **Tunnels**: A scaled version of Example 1, shown in Figure 2. Details on each problem, experimental setup, and algorithm implementation are in the Appendix. For all algorithms except CGCP-CL, solve time is limited to 300 seconds and online action selection to 0.05 seconds. For CGCP-CL, 300 seconds was given to re-compute each action. We report the mean discounted cumulative reward and cost, and constraint violation rate in Table 1. The constraint violation rate is the fraction of trials in which $d(h_t)$ becomes negative, which means Eq. (11) is violated.

In all environments, ARCS found admissible policies ($k = \infty$). In contrast, CGCP, Exp-Gradient, CPBVI and CPBVI-D only guarantees an admissible horizon of $k = 1$, since the C-POMDP constraint is only at the initial belief. CGCP-CL may have a closed-loop admissible horizon greater than 1, but does not provide guarantees, as indicated in the violation rate.

The benchmarking results show that the policies computed for ARCS generally achieve competitive cumulative reward to policies computed for C-POMDP, without any constraint violations and thus no pathological behavior. ARCS also

[3]Our code is open sourced at https://github.com/CU-ADCL/RC-PBVI.jl

| Env. | Algorithm | Violation Rate | Reward | Cost |
|---|---|---|---|---|
| CE

($\hat{c} = 5$) | CGCP | 0.51 | **12.00** | 5.19 |
| | CGCP-CL | **0.00** | 6.12 | 3.25 |
| | Exp-Gradient | 0.49 | 11.87 | 4.98 |
| | CPBVI | **0.00** | 8.39 | 4.38 |
| | CPBVI-D | **0.00** | 6.10 | 3.54 |
| | Ours | **0.00** | **10.00** | 5.00 |
| C-Tiger

($\hat{c} = 3$) | CGCP | 0.75 | $-1.69$ | 3.00 |
| | CGCP-CL | 0.14 | $-2.98$ | 2.93 |
| | Exp-Gradient | 1.0 | **1.81** | 3.22 |
| | CPBVI | 0.15 | $-11.11$ | 2.58 |
| | CPBVI-D | 0.09 | $-9.49$ | 2.76 |
| | Ours | **0.00** | **$-5.75$** | 2.98 |
| CRS(4,4)

($\hat{c} = 1$) | CGCP | 0.51 | **10.43** | 0.51 |
| | CGCP-CL | 0.78 | 1.68 | 0.72 |
| | Exp-Gradient | 0.30 | 10.38 | 0.92 |
| | CPBVI | **0.00** | $-0.40$ | 0.52 |
| | CPBVI-D | **0.00** | 0.64 | 0.47 |
| | Ours | **0.00** | **6.52** | 0.52 |
| CRS(5,7)

($\hat{c} = 1$) | CGCP | 0.41 | **11.98** | 1.00 |
| | CL-CGCP | 0.18 | 9.64 | 0.99 |
| | Exp-Gradient | 0.30 | 11.90 | 1.31 |
| | CPBVI | **0.00** | 0.00 | 0.00 |
| | CPBVI-D | **0.00** | 0.00 | 0.00 |
| | Ours | **0.00** | **11.77** | 0.95 |
| CRS(7,8)

($\hat{c} = 1$) | CGCP | 0.36 | 10.78 | 0.945 |
| | CL-CGCP | 0.20 | **11.17** | 0.931 |
| | EXP-Gradient | 0.32 | 10.03 | 1.15 |
| | CPBVI | **0.00** | 0.0 | 0.0 |
| | CPBVI-D | **0.00** | 0.0 | 0.0 |
| | Ours | **0.00** | 6.61 | 0.960 |
| Tunnels

($\hat{c} = 1$) | CGCP | 0.50 | 1.61 | 1.01 |
| | CL-CGCP | 0.31 | 1.22 | 0.68 |
| | Exp-Gradient | 0.48 | 1.35 | 0.82 |
| | CPBVI | 0.90 | **1.92** | 1.62 |
| | CPBVI-D | 0.89 | **1.92** | 1.57 |
| | Ours | **0.00** | **1.03** | 0.44 |

Table 1: Results for benchmarks. We report the mean for each metric. We bold the best violation rates in **black**, the highest reward with violation rate greater than 0 in blue, and the highest reward with 0 violation rate in green. Standard error of the mean, and problem parameters can be found in the appendix.

generally performs better in all metrics than CPBVI and CPBVI-D, both of which could not search the problem space sufficiently to find good solutions in large RC-POMDPs.

Although the C-POMDP policies generally satisfy the C-POMDP expected cost constraints, the prevalence of high violation rates of C-POMDP policies across the environments strongly suggests that the manifestation of the *stochastic self-destruction* in C-POMDPs is not an exceptional phenomenon, but intrinsic to the C-POMDP problem formulation. This behavior is illustrated in the Tunnels problem, shown in Figure 2. CGCP (in blue) decides to traverse tunnel $A$ 51% of the time even when it observes that $A$ is rocky, and traverses tunnel $B$ 49% of the time. In contrast, ARCS never traverses tunnel $A$, since such a policy is inadmissible. Instead, it traverses $B$ or $C$ depending on observation of

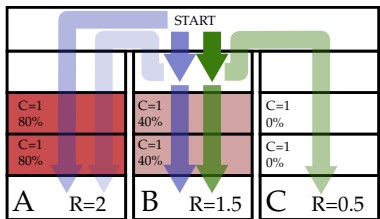

Figure 2: Tunnels. There is a cost of 1 for rock traversal (red regions) and 0.5 for backtracking. Trajectories from CGCP (blue) and ARCS (green) are displayed, with opacity approximately proportional to frequency of trajectories.

rocks in tunnel $B$, to maximize rewards while remaining admissible.

Finally, the closed-loop inconsistency of C-POMDP policies is evident when comparing open loop CGCP with closed loop CGCP-CL. In most cases (all except CRS(7,8)), the cumulative reward is decreased when going from CGCP to CGCP-CL, sometimes drastically. The violation rate also decreases, but not to 0, suggesting that planning with C-POMDPs instead of RC-POMDPs can lead to myopic behavior that cannot be addressed by re-planning. As seen in CE and both CRS, CGCP-CL attains lower reward than ARCS while still having constraint violations. Therefore, even for closed-loop planning, RC-POMDP can be more advantageous than C-POMDP.

**UNCONSTRAINED POMDP PROBLEMS**

Next, we additionally evaluate how well the RC-PODMP framework and our proposed algorithm performs for problems that have reduced constraints, so as to become equivalent to an unconstrained POMDP. We evaluate ARCS (RC-POMDP), CGCP (C-POMDP algorithm) and SARSOP (unconstrained POMDP algorithm) for the same benchmark problems with very high constraint thresholds $\hat{c} = 1000$.

For these problems, all policies are admissible, and our algorithm is guaranteed to asymptotically converge to the optimal solution. However, since our algorithm needs to keep track of admissible cost values, we utilize a policy tree representation. This representation is less efficient than the $\alpha$-vector policy representation used in SARSOP and CGCP, which allow value improvements at a belief state to directly improve values at other belief states.

Table 2 reports the lower bound reward and upper bound costs computed by each algorithm, with a time limit of $300s$. As seen in Table 2, our algorithm performs similar to CGCP and the unconstrained POMDP algorithm SARSOP for most smaller problems. The C-Tiger problem benefits greatly from the $\alpha$-vector representation, since the optimal policy repeatedly cycles among a small set of belief states (which our algorithm considers different augmented

belief-admissible cost states). For slightly larger problems (CRS(5,7)), the efficient $\alpha$-vector representation and other heuristics of SARSOP (which CGCP takes advantage of, since it repeatedly calls SARSOP) enables much faster convergence than the policy tree-based method of our approach. Nonetheless, as time is increased, our algorithm slowly improves its values.

| Env. | Algorithm | Reward | Cost |
|---|---|---|---|
| CE | SARSOP (POMDP) | **12.0** | - |
| | CGCP (C-POMDP) | **12.0** | 5.0 |
| | Ours (RC-POMDP) | **12.0** | 5.0 |
| C-Tiger | SARSOP (POMDP) | **1.93** | - |
| | CGCP (C-POMDP) | 1.90 | 3.2 |
| | Ours (RC-POMDP) | -1.4 | 3.2 |
| Tunnels | SARSOP (POMDP) | **1.92** | - |
| | CGCP (C-POMDP) | **1.92** | 1.6 |
| | Ours (RC-POMDP) | **1.92** | 1.6 |
| CRS(4,4) | SARSOP (POMDP) | **16.9** | - |
| | CGCP (C-POMDP) | **16.9** | 2.4 |
| | Ours (RC-POMDP) | **16.9** | 2.2 |
| CRS(5,7) | SARSOP (POMDP) | **23.9** | - |
| | CGCP (C-POMDP) | 14.8 | 3.6 |
| | Ours (RC-POMDP) | 14.9 | 2.1 |
| CRS(5,7) 1000s | SARSOP (POMDP) | **24.0** | - |
| | CGCP (C-POMDP) | 24.0 | 4.5 |
| | Ours (RC-POMDP) | 15.3 | 2.2 |

Table 2: Results for computed policy under-approximation (lower bound for reward values and upper bounds for cost values), best reward values in bold. SARSOP only considers reward value as an unconstrained POMDP algorithm.

# 7 CONCLUSION AND FUTURE WORK

We introduce and analyze the *stochastic self-destruction* behavior of C-POMDP policies, and show C-POMDPs may not exhibit optimal substructure. We propose a new formulation, RC-POMDPs, and present an algorithm for RC-POMDPs. Results show that C-POMDP policies exhibit unintuitive behavior not present in RC-POMDP policies, and our algorithm effectively computes policies for RC-POMDPs. We believe RC-POMDPs are an alternate formulation that can be more desirable for some applications.

For future work, we plan to study study what models exhibit strong cases of stochastic self-destruction, and develop more metrics that signal stochastic self-destruction. Additionally, we plan to analyze classes (or conditions) of RC-POMDPs that are approximable and designing algorithms that converge for such cases.

Further, our offline policy tree search algorithm can benefit from better policy search heuristics and more efficient policy representations (e.g. finite state controllers). We also plan to explore other approaches, such as searching for finite state controllers directly [Wray and Czuprynski, 2022] and online tree search approximations [Lee et al., 2018].

Finally, we have shown that RC-POMDPs can provide more desirable policies than C-POMDPs, but the cost constraints remain on expectation. For some applications, probabilistic or risk measure constraints may be more desirable than expectation constraints. These formulations also benefit from the recursive constraints that we propose for RC-POMDPs.

## Acknowledgements

Part of this research was carried out at the Jet Propulsion Laboratory, California Institute of Technology, under a contract with the National Aeronautics and Space Administration (80NM0018D0004).

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

# Recursively-Constrained Partially Observable Markov Decision Processes (Supplementary Material)

**Qi Heng Ho**[1]    **Tyler Becker**[1]    **Benjamin Kraske**[1]    **Zakariya Laouar**[1]    **Martin S. Feather**[2]    **Federico Rossi**[2]

**Morteza Lahijanian**[1]                                   **Zachary Sunberg**[1]

[1]Department of Aerospace Engineering Sciences, University of Colorado Boulder, Boulder, Colorado, USA
[2]Jet Propulsion Laboratory, California Institute of Technology, Pasadena, California, USA

## A  PROOFS OF THEOREM 1

*Proof.* Consider an optimal (stochastic) policy $\pi^*$. Consider a $h_t$ reachable from $b_0$ by $\pi^*$, where $\pi^*$ has randomization (at least two actions have probabilities in $(0, 1)$). Then, $\pi^*$ can be equivalently represented as a mixture over $n \leq |A|$ policies that deterministically selects a unique action at $h_t$ but is the same (stochastic) policy as $\pi^*$ everywhere else. That is, $\pi^*$ is a mixed policy over the set $\vec{\pi}^* = \{\pi_1, \cdots, \pi_n\}$ of $n$ policies that have a deterministic action at $h_t$ and $\pi_i = \pi_j = \pi^*$ at every other history. Let $w_i$ represent the non-zero probability of choosing $\pi_i$ at $h_t$. Then, $V_R^{\vec{\pi}^*}(h_t) = \sum_{i=1}^n w_i V_R^{\pi_i}(b_t)$ and $V_C^{\vec{\pi}^*}(h_t) = \sum_{i=1}^n w_i V_C^{\pi_i}(b_t)$.

We show that all the policies $\pi_i \in \vec{\pi}^*$ must be admissible in order for $\vec{\pi}^*$ to be admissible. Suppose there exists an inadmissible policy $\pi_j \in \vec{\pi}^*$. That is, there exists a history $h_f, f \geq t$, s.t. Eq. (11) or (14) is violated by $\pi_j$.

For $f > t$, $h_f$ is only reachable by taking action $\pi_j(h_t)$ at $h_t$, since each policy in $\vec{\pi}^*$ takes a different action at $h_t$, and so their reachable history spaces are different. Only the inadmissible $\pi_j$ is executed from $h_f$ when it is reached probablistically, and $W(h_t) + \gamma^t Q_C^{\pi_j}(b_f, \pi_j(h_f)) \not\leq \hat{c}$. This means that Eq.(11) is violated at depth $f$, so $\vec{\pi}^*$ is inadmissible, which is a contradiction.

Similarly, for $f = t$, if $V_C^{\pi_j}(b_t) \not\leq d(h_t)$ (Eq. (14) are violated),

$$\exists h_{t+1} \text{ s.t.} V_C^{\pi_j}(b_{t+1}) \not\leq d(h_{t+1}),$$

since

$$[\forall h_{t+1}, V_C^{\pi_j}(b_{t+1}) \leq d(h_{t+1})] \implies V_C^{\pi_j}(b_t) \leq d(h_t).$$

This can be seen by rearranging Eq. (13) and

$$V_C^{\pi_j}(b_t) = C(b_t, \pi_j(h_t)) + \gamma \mathbb{E}[V_C^{\pi_j}(b_{t+1})]$$

. That is, Eq. (14) is violated at time step $t + 1$, so $\vec{\pi}^*$ is inadmissible, which is again a contradiction. Hence, each $\pi_i \in \vec{\pi}^*$ is admissible.

Since $\max_i V_R^{\pi_i}(b_t) \geq \sum_{i=1}^n w_i V_R^{\pi_i}(b_t)$, determinism at $h_t$ is sufficient, i.e., obtain a new $\pi^* = \arg\max_{\pi_i}(V_R^{\pi_i}(b_t))$ with one less randomization. Repeating the same process for all histories reachable from $b_0$ by $\pi^*$ with randomization obtains a deterministic optimal policy. □

## B  PROOF OF THEOREM 2

*Proof.* Let $\mathbf{a}'$ denote the action set computed during the Bellman backup operation on $V^{\pi^*}(\bar{b}_t)$ to obtain $V^{\pi'}(\bar{b}_t)$.

$$\mathbf{a}' = \arg\max_{a \in A} \left[ Q_R^{\pi^*}(\bar{b}_t, a) \mid Q_C^{\pi^*}(\bar{b}_t, a) \leq d(h_t) \right]$$

We first show that $V_R^{\pi*}(\bar{b}_t) = V_R^{\pi'}(\bar{b}_t) \ \forall \bar{b}_t$, i.e., the optimality after a Bellman backup operation is preserved.

Consider any $\bar{b}_t$. By optimality of $\pi^*$, Suppose $\mathbf{a} \neq \emptyset$,

$$V^{\pi'}(\bar{b}_t) = \max_{a \in A} \left[ Q_R^\pi(\bar{b}_t, a) \mid Q_C^\pi(\bar{b}_t, a) \le d(h_t) \right]$$
$$V^{\pi'}(\bar{b}_t) > V^{\pi^*}(\bar{b}_t) \implies \pi^* \text{ is not optimal}$$

which is a contradiction, so $V^{\pi'}(\bar{b}_t) \le V^{\pi^*}(\bar{b}_t)$. However, we have that

$$V_R^{\pi'}(\bar{b}_t) = \max_a [Q_R^\pi(\bar{b}_t, a)]$$
$$\implies V^{\pi'} \ge V_R^{\pi^*}(\bar{b}_t)$$
$$\implies V^{\pi'} = V_R^{\pi^*}(\bar{b}_t).$$

Next, we have that if $\mathbf{a} = \emptyset$, from Eq. (15)

$$V_R^{\pi'}(\bar{b}_t) = V_R^{\pi^*}(\bar{b}_t)$$

Therefore, $V_R^{\pi*}(\bar{b}_t) = V_R^{\pi'}(\bar{b}_t) \ \forall \bar{b}_t$.

Now, we show that $V_C^{\pi'}(b_t) \le d(h_t), \ \forall \bar{b}_t \in \text{REACH}^{\pi'}(\bar{b}_0)$, i.e., the admissibility of the optimal policy after a Bellman backup operation is preserved.

Let $\bar{b}_t \in \text{REACH}^{\pi'}(\bar{b}_0), t \in \mathbb{N}$ be the first augmented belief state from $b_0$ in a belief trajectory such that $V_C^{\pi'}(\bar{b}_t)$ does not satisfy Eq. 14, i.e., $\pi'$ satisfies Eq. 14 $\forall \tau < t$, and $V_C^{\pi'}(\bar{b}_t) \not\le d(h)$.

$$V_C^{\pi'}(\bar{b}_t) \not\le d(h_t) \implies \mathbf{a}' = \emptyset$$

Therefore,

$$V_R^{\pi^*}(\bar{b}_t) = V_R^{\pi'}(\bar{b}_t),$$
$$V_C^{\pi'}(\bar{b}_t) = V_C^{\pi^*}(\bar{b}_t) = (\infty, \dots, \infty).$$

Consider the augmented belief state $\bar{b}_{t-1}$ that transitions to $\bar{b}_t$ under some action $a'_{t-1} = \pi'(\bar{b}_{t-1})$ and observation $o_{t-1}$.

$$Q_C^{\pi^*}(\bar{b}_{t-1}, a'_{t-1}) = C(b, a) + \mathbb{E}[V_R^{\pi^*}(\bar{b}')]$$
$$= C(b, a) + (\infty, \dots, \infty)$$
$$= (\infty, \dots, \infty) > d(h_{t-1}).$$

Note that $d(h_{t-1}) \le \frac{\hat{c}}{\gamma^{t-1}}$ is finite for finite $t-1$. So $a'_{t-1} \notin \arg\max_{a \in A} \left[ Q_R^{\pi^*}(\bar{b}_{t-1}, a) \mid Q_C^{\pi^*}(\bar{b}_{t-1}, a) \le d(h_{t-1}) \right]$ and therefore $\bar{b}_t$ cannot be reachable under $\pi'$. By induction, we have that $\forall \bar{b}_t \in \text{REACH}^{\pi'}(\bar{b}_0)$,

$$V_C^{\pi'}(\bar{b}_t) \le d(h).$$

$\square$

## B.1 PROOF OF THEOREM 3

*Proof.* Suppose that $\pi^0$ is well behaved.

Denote $B_{inadmiss} = \{\bar{b} \mid \Phi = \emptyset\}$. Then, for all $\bar{b} = (b, d) \in B_{inadmiss}$, $V_C^{\pi^0}(\bar{b}) \not\le d(h_t)$. A lack of admissible policy implies that there are no actions from $\bar{b}_t$ that leads to admissibility. This implies that $\forall a \in A$, there exists at least 1 successor augmented belief state $\bar{b}'$, such that $\Phi(\bar{b}') = \emptyset$. Therefore,

$$\mathbf{a} = \arg\max_{a \in A} \left[ Q_R^\pi(\bar{b}, a) \mid Q_C^\pi(\bar{b}, a) \le d \right] = \emptyset$$
$$\mathbb{B}[V^{\pi^0}](\bar{b}) = (V_R^{\pi^0}(\bar{b}_t), (\infty, \dots, \infty)).$$

Therefore, we have that for all $\bar{b} \in B_{inadmiss}$,

$$\mathbb{B}^n[V^{\pi^0}](\bar{b}) = (V_R^{\pi^0}(\bar{b}_t), (\infty, \ldots, \infty)) \ \ \forall n.$$

Next, denote $B_{admiss} = \{\bar{b}_t \mid \Phi(\bar{b}_t) \neq \emptyset\}$. Consider any $\bar{b} \in B_{admiss}$, $V_R^{\pi^0}(\bar{b}) \in \mathbb{R}$ and $V_C^{\pi^0}(\bar{b}) \leq d(h_t)$. There must exist at least 1 action that is part of an admissible policy, i.e., $\mathbf{a} \neq \emptyset$. Therefore, at $\bar{b}$

$$\mathbb{B}[V^\pi](\bar{b}) = \left( Q_R^\pi(\bar{b}, a), Q_C^\pi(\bar{b}, a) \right), a \in \mathbf{a}$$

Since any $a \notin \mathbf{a}$ are inadmissible and will not be selected during the Bellman backup operation, we can exclude them from the set of actions without loss of generality. Denote $A'(\bar{b}) = \mathbf{a}$ as the set of actions that may be selected at $\bar{b}$. Then, for all $\bar{b} = (b, d) \in B_{admiss}$,

$$\mathbf{a}' = \arg\max_{a \in A'(\bar{b})} \left[ Q_R^\pi(\bar{b}, a) \right]$$
$$\mathbb{B}[V^{\pi^0}](\bar{b}) = (Q_R^{\pi^0}((\bar{b}), a), Q_C^{\pi^0}(\bar{b}, a)), a \in \mathbf{a}'.$$

Consider $V^{\pi^1} = \mathbb{B}[V^{\pi^0}](\bar{b})$. We have that

$$V_R^{\pi^1}(\bar{b}) \geq V_R^{\pi^0}(\bar{b}) \text{ and } V_C^{\pi^1}(\bar{b}) \leq d.$$

By induction, $V_C^{\pi^n}(\bar{b}) \leq d \ \forall \bar{b} \in B_{admiss}$, so the policy $\pi^n$ remains admissible after applying $\mathbb{B}$. Therefore, we can write

$$\mathbb{B}_R[V_R^\pi](\bar{b}) = \max_{a \in A'(\bar{b})} \left[ Q_R^\pi(\bar{b}, a) \right], \forall \bar{b} \in B_{admiss}.$$

Note that this is the standard Bellman operator for an unconstrained discounted-sum POMDP over the set of admissible augmented belief states. From the results of the Bellman operator for a POMDP [Hauskrecht, 1997], $\mathbb{B}_R$ is a contraction mapping and has a single, unique fixed point, i.e., for an optimal $\pi^*$, $\mathbb{B}_R[V_R^{\pi^*}](\bar{b}) = V_R^{\pi^*}(\bar{b}) \ \ \forall \bar{b} \in B_{admiss}$. Since $V_R^{\pi^n}(\bar{b}) = V_R^{\pi^0}(\bar{b}) \ \ \forall \bar{b} \in B_{inadmiss}, \forall n$, we have that

$$\mathbb{B}^n[(V_R^{\pi^0}, V_C^{\pi^0})] \to (V_R^{\pi^*}, V_C^{\pi^n}) \text{ as } n \to \infty.$$

$\square$

## C ARCS PSEUDOCODE

Algorithm 2: Sampling of nodes for backup.

---

**Global variables**: $\mathcal{M}, T, \Gamma_{c_{min}}$

Let $\gamma = \mathcal{M}.P.\gamma$

SAMPLE($\epsilon$).

1: $L \leftarrow T.v_0.\underline{V}_R$.
2: $U \leftarrow L + \epsilon$.
3: **if** $rand() < 0.5$ **then**
4:     SampleHeu($T.v_0, L, U, \epsilon_t, \gamma, 1$).
5: **else**
6:     SampleRandom($T.v_0, \gamma$).
7: **end if**
8: **return** sampled nodes.

SampleHeu($v, L, U, \epsilon, \gamma, t$).

1: Let $\hat{V}$ be the predicted value of $v.V_R^*$.
2: **if** $\hat{V} \leq L$ and $v.\overline{V}_R \leq max\{U, \underline{V}_R(v.b) + \epsilon\gamma^{-t}\}$ **then**
3:     **return**.
4: **else**
5:     $Q \leftarrow \max_a \underline{Q}_R(v.b, a)$.
6:     $\overline{L}' \leftarrow \max\{\overline{L}, Q\}$.
7:     $U' \leftarrow \max\{U, \underline{Q} + \gamma^{-t}\epsilon\}$.
8:     $a' \leftarrow \arg\max_a\{v.\overline{Q}_R(a) \mid v.\underline{Q}_C(a) \leq v.d\}$.
9:     **if** $a' = \emptyset$ **then**
10:         **return**.
11:     **end if**
12:     $o' \leftarrow \arg\max_o[p(o|b, a')(v'.\overline{V}_R - v'.\underline{V}_R - \epsilon\gamma^{-t})]$.
13:     Compute $L_t$ such that $L' = R(v.b, a') +$
    $\gamma(p(o'|b, a')L_t + \sum_{o \neq o'} p(o|v.b, a')v'.\underline{V}_R)$.
14:     Compute $U_t$ such that $U' = R(v.b, a') +$
    $\gamma(p(o'|b, a')U_t + \sum_{o \neq o'} p(o|v.b, a')v'.\overline{V}_R)$.
15:     $v' \leftarrow$ SuccessorNode($v, a', o'$).
16:     $T \leftarrow v'$.
17:     SampleHeu($v', L_t, U_t, \epsilon, t+1$).
18: **end if**

SampleRandom($v, \gamma$)

1: $a \leftarrow rand_a\{a \in A\}$.
2: $o \leftarrow rand_o\{o \in O\}$.
3: $v' \leftarrow$ SuccessorNode($v, a, o$).
4: $T \leftarrow v'$.
5: **if** new node is added to $T$ **then**
6:     **return**.
7: **else**
8:     SampleRandom($v', \gamma$).
9: **end if**

---

# D  PROOF OF LEMMA 1

*Proof.* Given a maximum one-step cost $C_{max}$ for $\pi_c^{min}$ and a non-negative admissible cost $v.d$, a lower bound on the admissible horizon can be obtained as follows. The $k$-step admissible horizon from each leaf node $v$ is the largest $k$ such that

$$\sum_{\tau=0}^{k-1} \gamma^\tau C_{max} \leq v.d.$$

---

Algorithm 3: Compute Successor Node.

---

**Global variables**: $\mathcal{M}, T, \Gamma_{c_{min}}$
Let $\gamma = \mathcal{M}.P.\gamma$.
SuccessorNode$(v, a, o)$

 1: **if** $T.child(v, a, o) \notin T$ **then**
 2:    $b' \leftarrow BeliefUpdate(v.b, a, o)$ using Eq. (1).
 3:    $d' \leftarrow \frac{1}{\gamma}(v.d - C(v.b, a))$.
 4:    Initialize lower bound on $k'$ using Eq. (22).
 5:    $(\alpha_r, \alpha_c) \leftarrow \arg\min_{(\alpha_r, \alpha_c) \in \Gamma_{c_{min}}} \alpha_r^T b'$.
 6:    $\underline{V}_R \leftarrow \alpha_r^T b'$.
 7:    $\overline{V}_C \leftarrow \alpha_c^T b'$.
 8:    $\overline{V}_R \leftarrow \overline{V}_R(b')$ with Fast Informed Bound (maximizing rewards).
 9:    $\underline{V}_C \leftarrow \underline{V}_C(b')$ with Fast Informed Bound (minimizing costs).
10:    $\overline{Q}_R \leftarrow \emptyset$.
11:    $\underline{Q}_R \leftarrow \emptyset$.
12:    $\overline{Q}_C \leftarrow \emptyset$.
13:    $\underline{Q}_C \leftarrow \emptyset$.
14:    $v' \leftarrow (b', d', k', \overline{V}_R, \underline{V}_R, \overline{V}_C, \overline{Q}_R, \underline{Q}_R, \overline{Q}_C)$.
15: **else**
16:    $v' \leftarrow T.child(v, a, o)$.
17: **end if**
18: **return** $v$.

---

The LHS is a finite geometric series:

$$\sum_{\tau=0}^{k-1} \gamma^\tau C_{max} = C_{max}\left(\frac{1 - \gamma^k}{1 - \gamma}\right) \leq v.d. \tag{23}$$

Hence, the largest integer $k$ that satisfies Eq. (23) is

$$k = \left\lfloor \log\left(1 - \left(\frac{d(h_t)}{C_{max}}\right) \cdot (1 - \gamma)\right) / \log(\gamma) \right\rfloor.$$

Also, we obtain the $\infty$-admissibility condition on $\pi_c^{min}$ by setting $k = \infty$ in Eq. (23):

$$\frac{C_{max}}{1 - \gamma} \leq v.d.$$

Next, when $v.\overline{V}_C^{\pi_c^{min}} = 0 \leq v.d$, since costs are non-negative, the minimum cost policy obtains 0 cost at every future belief, and hence $k = \infty$.

Finally, when admissible cost is negative, the constraints are trivially violated, and hence $k = 0$. $\qquad\square$

## E   PROOF OF PROPOSITION 3

*Proof.* During search, the pruning criteria prunes policies according to four cases. We show that these four cases only prunes sub-optimal policies and inadmissible policies.

1. $v.\underline{V}_C > v.d$.
   It is easy to see that if $v.\underline{V}_C > v.d$, it is guaranteed that no admissible policies exist from $v$, so we can prune $v$ and its subtree.

2. At node $v$, actions $a$ and $a'$ are compared. Specifically, $v.\overline{Q}_R(a)$ is compared with $v.\underline{Q}_R(a')$, if $k(v, a') = \infty$ (the policy from taking $a'$ is admissible). The node-action $(v, a)$ is pruned if $v.\overline{Q}_R(a) < v.\underline{Q}_R(a')$.

---

**Algorithm 4: Perform backup at a node**

---

**Global variables**: $\mathcal{M}, T, \Gamma_{c_{min}}$

Let $\gamma = \mathcal{M}.P.\gamma$

BACKUP $(v)$

1:   Initialize $\vec{k}$ of size $|A|$
2:   **for all** $a \in A$ **do**
3:     $v.\overline{Q}_R(a) \leftarrow R(v.b, a) + \gamma \mathbb{E}[v'.\overline{V}_R]$
4:     $v.\overline{Q}_C(a) \leftarrow C(v.b, a) + \gamma \mathbb{E}[v'.\overline{V}_C]$
5:     $v.\underline{Q}_R(a) \leftarrow R(v.b, a) + \gamma \mathbb{E}[v'.\underline{V}_R]$
6:     $v.\underline{Q}_C(a) \leftarrow C(v.b, a) + \gamma \mathbb{E}[v'.\underline{V}_C]$
7:     $\vec{k}[a] \leftarrow \min_{v'} v'.k$
8:   **end for**
9:   $a \leftarrow \arg\max_a \{v.\underline{Q}_R(a) \mid \overline{Q}_C(a) \leq v.d\}$
10:   **if** $a \neq \emptyset$ **then**
11:     $v.\underline{V}_R \leftarrow v.\underline{Q}_R(a)$
12:     $v.\overline{V}_C \leftarrow v.\overline{Q}_C(a)$
13:     $v.k \leftarrow \vec{k}[a] + 1$
14:   **else**
15:     $a \leftarrow \arg\min_a \{v.\overline{Q}_C(a)\}$
16:     $v.\underline{V}_R \leftarrow v.\underline{Q}_R(a)$
17:     $v.\overline{V}_C \leftarrow v.\overline{Q}_C(a)$
18:     $v.k \leftarrow 0$
19:   **end if**
20:   $a \leftarrow \arg\max_a \{v.\overline{Q}_R(a) \mid \underline{Q}_C(a) \leq v.d\}$
21:   **if** $a \neq \emptyset$ **then**
22:     $v.\overline{V}_R \leftarrow v.\overline{Q}_R(a)$
23:     $v.\underline{V}_C \leftarrow v.\underline{Q}_C(a)$
24:   **else**
25:     $v.\overline{V}_R \leftarrow -\infty$
26:     $v.\underline{V}_C \leftarrow \infty$
27:     $v.\underline{V}_R \leftarrow -\infty$
28:     $v.\overline{V}_C \leftarrow \infty$
29:   **end if**

---

There are two cases to consider: (i) $k(v, a) = \infty$ and (ii) $k(v, a) < \infty$. In case (i), $k(v, a) = \infty$. $v.\overline{Q}_R(a)$ is a valid upper bound of the Q reward-value of taking action $a$ as a consequence of Lemma 2. In case (ii), $k(v, a) < \infty$, $v.\overline{Q}_R(a)$ is also a valid upper bound of the Q reward-value of taking action $a$. For a node $v$ with $\bar{b} = (v.b, v.d)$, We show that $v.\overline{V}_R$ is an upper bound on $V_R^*(\bar{b})$. That is, if $v.k \leq \infty$, we have that for an optimal policy starting from $\bar{b} = (v.b, v.d)$ with optimal reward-value $V_R^*(\bar{b})$ and cost-value $V_C^*(\bar{b})$,

$$V_R^*(\bar{b}) \leq v.\overline{V}_R \text{ and } v.\underline{V}_C \leq V_C^*(\bar{b}). \tag{24}$$

This can be seen by noting that the BACKUP step performs an RC-POMDP Bellman backup. If the policy is in fact admissible (since $v.k$ is an underestimate), then the results from Lemma 2 hold. If the policy is not admissible, the optimal reward-value of an admissible policy from that node cannot be higher than $v.\overline{V}_R$ (and the optimal cost-value cannot be higher than $v.\underline{V}_C$), since an admissible policy satisfies more constraints than a finite $k$-admissible one. In both cases, $a'$ is strictly a better action than $a$, so taking action $a$ at node $v$ cannot be part of an optimal policy.

3. If all node-actions $(v, a)$ are pruned, $v$ is also pruned.
   No actions are admissible from $v$ so it is inadmissible.

4. $(v, a)$ is pruned if any successor node from taking action $a$ is pruned.
   A successor node is pruned (case 1 or 3), so this policy is not admissible.

$\square$

---

Algorithm 5: Prune nodes and node-actions from $v$

---

**Global variables**: $\mathcal{M}, T, \Gamma_{c_{min}}$

PRUNE$(v)$

 1: **for all** $v \in B_{sam}$ **do**
 2:    **if** $v.\underline{V}_C > v.d$ **then**
 3:       Prune $v$.
 4:    **end if**
 5:    **if all** node-actions $(v, a)$ are pruned **then**
 6:       Prune $v$.
 7:    **end if**
 8:    Initialize $\vec{k}$ of size $|A|$.
 9:    **for all** $a \in A$ **do**
10:       $\vec{k}[a] \leftarrow \min v'.k$.
11:    **end for**
12:    **for all** $a, a' \in A$ **do**
13:       **if** any child of $(v, a)$ is pruned **then**
14:          Prune node-action $(v, a)$.
15:       **end if**
16:       **if** $\vec{k}[a'] = \infty, v.\overline{Q}_R(a) < v.\underline{Q}_R(a')$ **then**
17:          Prune node-action $(v, a)$.
18:       **end if**
19:    **end for**
20: **end for**

---

# F   PROOF OF LEMMA 2

*Proof.* The proof relies on the result of Theorem 1. that for admissibility constraint $k = \infty$, deterministic policies are sufficient for optimality. That is, it is sufficient to provide upper and lower bounds over deterministic policies.

We first show that the initial bounds $\overline{V}_R, \underline{V}_R, \overline{V}_C, \underline{V}_C, k$ for a new (leaf) node $v$ (Algorithm 3) are true bounds on the optimal policy.

$\overline{V}_R$ and $\underline{V}_C$ are initialized with the Fast Informed Bound with the unconstrained POMDP problem, separately for reward maximization and cost minimization. The Fast Informed Bound provides valid upper bounds on reward-value (and lower bounds on cost-value of a cost-minimization policy, which in turn is a lower bound on cost-value for an optimal RC-POMDP policy) Hauskrecht [2000]. The upper bound on the optimal reward-value for the reward-maximization unconstrained POMDP problem is also an upper bound on the optimal reward-value for an RC-POMDP which has additional constraints. Similarly, the lower bound on the optimal cost-value for the cost-minimization unconstrained POMDP problem is also a lower bound on the cost-value of an optimal RC-POMDP policy which has additional constraints.

$\underline{V}_R$ and $\overline{V}_C$ are computed using the minimum cost policy $\Gamma_{c_{min}}$. This minimum cost policy is an alpha-vector policy, which is an upper bound on the cost-value function Hauskrecht [2000], so $\overline{V}_C$ is an upper bound on the cost-value when following the minimum cost policy. It can also be seen that the value $\underline{V}_R$ from following the same policy is a valid bound on the optimal reward-value function from that node. Finally, the admissible horizon guarantee $k$ is initialized using the results from Lemma 1, which is shown to be a lower bound on the true admissible horizon following the minimum cost policy.

Next, we show that performing the BACKUP step (Algorithm 4 maintains the validity of the bounds. Recall the condition of this lemma that admissible horizon guarantee $v.k = \infty$ for the node $v$. Thus, after the backup step, the admissible horizon guarantee remains at $v.k = \infty$. From the proof of Theorem 2, the RC-POMDP Bellman backup $\mathbb{B}$ satisfies Bellman's Principle of Optimality and is a contraction mapping within the space of admissible value functions (and hence policies).

Let $\overline{V}'_R, \underline{V}'_R, \overline{V}'_C, \underline{V}'_C$ be the value after the BACKUP step, which performs a Bellman backup $\mathbb{B}$ on $\overline{V}_R, \underline{V}_R, \overline{V}_C, \underline{V}_C$:

$$v.\overline{V}'_R = v.\mathbb{B}(\overline{V}_R),$$
$$v.\underline{V}'_R = v.\mathbb{B}(\underline{V}_R),$$
$$v.\overline{V}'_C = v.\mathbb{B}(\overline{V}_C),$$
$$v.\underline{V}'_C = v.\mathbb{B}(\underline{V}_C).$$

Since $\mathbb{B}$ is a contraction mapping within the space of admissible policies, we see that:

$$v.\mathbb{B}(\overline{V}_R) \leq v.\overline{V}_R,$$
$$v.\mathbb{B}(\overline{V}_C) \leq v.\overline{V}_C,$$
$$v.\mathbb{B}(\underline{V}_R) \geq v.\underline{V}_R,$$
$$v.\mathbb{B}(\underline{V}_C) \geq v.\underline{V}_C.$$

Therefore, for $v.k = \infty$, we have that for an optimal policy starting from $\bar{b} = (v.b, v.d)$ with optimal reward-value $V_R^*(\bar{b})$ and cost-value $V_C^*(\bar{b})$,

$$v.\underline{V}_R \leq V_R^*(\bar{b}) \leq v.\overline{V}_R$$
$$V_C^*(\bar{b}) \leq v.\overline{V}_C$$

$\square$

# G  PROOF OF THEOREM 4

*Proof.* There are two termination criteria for ARCS, of which both must be true before termination. ARCS terminates when (1) it finds an admissible policy, and (2) the policy is $\epsilon$-optimal, that is when $v_0.\overline{V}_R - v_0.\underline{V}_R \leq \epsilon$. We first discuss admissibility, then $\epsilon$-optimality.

(1) ARCS can terminate when it finds an admissible policy, i.e., $v_0.k = \infty$. ARCS finds an admissible policy when every leaf node $v_{leaf}$ under the policy satisfies (i) Eq. (20), or (ii) $V_C^\pi(b_t') = 0$.

We prove that this is a sound condition, i.e., if the (i) and (ii) hold for every leaf node $v_{leaf}$, the computed policy is indeed admissible. As proven in Lemma 1, the admissible horizon guarantee for a leaf node $v_{leaf}.k$ is a conservative under-approximation. Therefore, a leaf node with $v_{leaf}.k = \infty$ indeed means that we have found an admissible policy from $v_{leaf}$ (with $\Gamma_{c_{min}}$). Suppose all leaf nodes have $v_{leaf}.k = \infty$. The worst-case back-propagation of admissible horizon guarantee up the tree is sound, since a non-leaf node $v$ only has $v.k = \infty$ if all its leaf nodes have $k = \infty$ *and* Eq. (9) is satisfied at that node (Lines 9-18 in Algorithm 4). Therefore, if $v_0.k = \infty$, the policy is admissible, and ARCS can terminate if $v_0.k = \infty$.

(2) ARCS can terminate when the gap criterion at the root is satisfied, that is when $v_0.\overline{V}_R - v_0.\underline{V}_R \leq \epsilon$.

If $v_0.k = \infty$, the policy at $v_0.k$ is admissible, which implies every history-belief reachable under the policy tree is admissible. From Lemma 2, this implies that for all nodes, $\overline{V}_R, \underline{V}_R, \overline{V}_C, \underline{V}_C$ are valid bounds on the optimal value function. Thus, $v_0.\overline{V}_R$ and $v_0.\underline{V}_R$ are valid bounds on the optimal value function from $b_0$, and so, an $\epsilon$-optimal policy is indeed found.

Therefore, if ARCS terminates, the computed solution is an admissible $\epsilon$-optimal policy. $\square$

# H  EXPERIMENTAL EVALUATION

## H.1  IMPLEMENTATION DETAILS

The code for each algorithm implementation can be found in the attached supplementary material. Here, we detail parameters and implementation of the algorithms. For hyper-parameter tuning, we used the default parameters for ARCS. For the rest of the algorithms, the values of the hyper-parameters were chosen based empirical evaluations, and fixed for the experiments. For each environment, we used a maximum time step of 20 during evaluation. Except for the Tiger problem, all algorithms reached terminal states before 20 time steps in these problems or produced a policy which stayed still at 20 time steps.

### H.1.1 ARCS

We implemented ARCS as described. We used the Fast Informed Bound for the initialization of upper bound on reward value and lower bound on cost value. We used SARSOP for the computation of the minimum cost policy. We set the SARSOP hyperparameter $\kappa = 0.5$ for our experiments, the same value as Kurniawati et al. [2008]. We used a uniform randomization (0.5 probability) between heuristic sampling and random sampling during planning, and we leave an analysis on how the randomization weight may affect planning efficiency to future work.

### H.1.2 CGCP

We implemented CGCP and adapted it for discounted infinite horizon problems, using Alg. 5 in Walraven and Spaan [2018] as a basis. However, we use the discounted infinite horizon POMDP solver SARSOP Kurniawati et al. [2008] in place of a finite horizon PBVI. Our method of constructing policy graphs also differs, as the approach described is for finite horizon problems. We check for a return to beliefs previously visited under the policy in order to reduce the size of the graph. A maximum time of 300 seconds was used for CGCP. For each SARSOP iteration within CGCP, $\tau = 20$ seconds was given initially, while the solve time was incremented by $\tau^+ = 100$ seconds every time that the dual price $\lambda$ remained the same. Additionally, CGCP was limited to 100 iterations. In an effort to reduce computation time, policy graph evaluation and SARSOP search was limited to depth 20 (the same as the monte carlo evaluation depth) in all domains except the RockSample domains, which were allowed unlimited depth.

For the Tunnels benchmark, 1000 monte carlo simulations to depth 20 were used in place of a policy graph to estimate the value of policies. This was due to the inability of the policy graphs to estimate the value of some infinite horizon POMDP solutions which do not lead to terminal states or beliefs which have already appeared in the tree.

### H.1.3 CGCP-CL

CGCP-CL uses the same parameters as CGCP, but re-plans at every time step.

### H.1.4 No-regret Learning Algorithm

We implemented the no-regret learning algorithm from [Kalagarla et al., 2022]. We used SARSOP as the unconstrained POMDP solver, and monte carlo simulations to estimate the value of policies.

### H.1.5 CPBVI

We implemented CPBVI based on Kim et al. [2011]. The algorithm generates a set of reachable beliefs $\mathcal{B}$ before performing iterations of approximate dynamic programming on the belief set. However, the paper did not include full details on belief set $\mathcal{B}$ generation and alpha-vector set $\Gamma$ initialization.

The paper cited Pineau et al. [2006] for their belief set description, and so we followed Pineau et al. [2006] by expanding $\mathcal{B}$ greedily towards achieving uniform density in the set of reachable beliefs. This is done by randomly simulating a step forward from a node in the tree, thereby generating candidate beliefs, and keeping the belief that is farthest from any belief already in the tree. We repeat this expansion until the desired number of beliefs have been added to the tree.

To address $\Gamma$ initialization, we adopted the *blind lower bound* approach. This approach represents the lower bound $\Gamma$ with a set of alpha-vectors corresponding to each action in $A$. Each alpha-vector is generated under the assumption that the same action is taken forever. To compute an alpha-vector corresponding to a given action, we first compute the *best-action worst-state (BAWS)* lower bound. This is done by evaluating the discounted reward obtained by taking the best action in the worst state forever. We can then update the *BAWS* alpha-vectors by performing value backups until convergence.

The CPBVI algorithm involves the computation of a linear program (LP) to obtain the best action at a given belief. One of the constraints asserts that the convex combination of cost alpha-vectors evaluated at a given belief $b$ must be equal to or less than the admissible cost $d$ associated with $b$, which is used in CPBVI's heuristic approach. However, if $d < 0$, the LP becomes infeasible. The case of $d < 0$ is possible since no pruning of beliefs is conducted. The paper did not provide details to account for this situation. To address this, if the LP is infeasible, we output the action with the lowest cost, akin to ARCS' minimum cost policy method when no policy is admissible.

### H.1.6 CPBVI-D

CPBVI computes stochastic policies. We modify CPBVI to only compute deterministic policies with the following details. Instead of solving the LP to generate a stochastic action, we solve the for the single highest value action subject to the cost-value constraint.

Although both CPBVI and CPBVI-D theoretically have performance insensitive to random seed initialization, both algorithms are sensitive to the number of belief parameter during planning. With too few beliefs selected for a problem, both algorithms cannot search the problem space sufficiently. With too many beliefs selected, the time taken for belief selection is too high for the moderately sized problems of CRS and Tunnels. Therefore, we tuned and chose a belief parameter of 30 that allows finding solutions in the planning time of $300s$. Note that even with a small number of beliefs of 30, CPBVI routinely overruns the planning time limit during its update step.

## H.2 ENVIRONMENT DETAILS

- **CE**: Simplified counterexample in Figure 1.
- **C-Tiger**: A constrained version of the Tiger POMDP Problem Kaelbling et al. [1998], with cost of 1 for the "listen" action.
- **CRS**: A constrained version of the RockSample problem Smith and Simmons [2004] as defined in Lee et al. [2018] with varying sizes and number of rocks.
- **Tunnels**: A scaled version of Example 1, shown in Fig. 2.

Except for the RockSample environment, our environments do not depend on randomness. For each RockSample environment of which rock location depends on randomness, we used the rng algorithm MersenneTwister with a fixed seed to generate the RockSample environments.

### H.2.1 Counterexample Problem

The counterexample POMDP in Figure 1 uses a discount of $\gamma = 1$. In the experiments, we used a discount factor of $\gamma = 1 - e^{-14}$ to approximate a discount of $\gamma = 1$. It is modeled as an RC-POMDP as follows.

States are enumerated as $\{s_1, s_2, s_3, s_4, s_5\}$ with actions following as $\{a_A, a_B\}$ and observations being noisy indicators for whether or not a state is rocky.

States $s_1$ and $s_2$ indicate whether cave 1 or cave 2 contains rocky terrain, respectively. Taking action $a_B$ circumvents the caves unilaterally incurring a cost of $5.0$ and transitioning to terminal state $s_5$. Taking action $a_B$ moves closer to the caves where $s_i$ deterministically transitions to $s_{i+2}$. In this transition, the agent is given an 85% accurate observation of the true state.

At this new observation position, the agent is given a choice to commit to one of two caves where $s_3$ indicates that cave 1 contains rocks and $s_4$ indicates that cave 2 contains rocks. Action $a_A$ moves through cave 1 and $a_B$ moves through cave 2. Moving through rocks incurs a cost of 10 while avoiding them incurs no cost. Taking action $a_A$ at this point, regardless of true state, gives a reward of 12. States $s_3$ and $s_4$ unilaterally transition to terminal state $s_5$.

### H.2.2 Tunnels Problem

The tunnels problem is modeled as an RC-POMDP as follows. As depicted in figure 2, the tunnels problem consists of a centralized starting hall that funnels into 3 separate tunnels. At the end of tunnels 1,2,and 3 lie rewards 2.0, 1.5 and 0.5 respectively. However, with high reward also comes high cost as tunnel 1 has a 80% probability of containing rocks and tunnel 2 has a 40% probability of containing rocks while tunnel 3 is always free of rocks. If present, the rocks fill 2 steps before the reward location at the end of a tunnel and a cost of 1 is incurred if the agent traverses over these rocks. Furthermore, a cost of 1 is incurred if the agent chooses to move backwards.

The only partial observability in the state is over whether or not rocks are present in tunnels 1 or 2. As the agent gets closer to the rocks, accuracy of observations indicating the presence of rocks increases.

## H.3   EXPERIMENT EVALUATION SETUP

We implemented each algorithm in Julia using tools from the POMDPs.jl framework Egorov et al. [2017], and all experiments were conducted single-threaded on a computer with two nominally 2.2 GHz Intel Xeon CPUs with 48 cores and 128 GB RAM. All experiments were conducted in Ubuntu 18.04.6 LTS. For all algorithms except CGCP-CL, solve time is limited to 300 seconds and online action selection to 0.05 seconds. For CGCP-CL, 300 seconds was given for each action (recomputed from scratch). We simulate each policy 1000 times, except for CGCP-CL, which is simulated 100 times due to the time taken for re-computation of the policy at each time step. The full results with the mean and standard error of the mean for each metric are shown in Table. 3.

| Environment | State/Action/Obs | Algorithm | Violation Rate | Cumulative Reward | Cumulative Cost |
|---|---|---|---|---|---|
| CE

$(\hat{c} = 5)$ | 5 / 2 / 2 | CGCP
CGCP-CL
CPBVI
CPBVI-D
EXP-Gradient
Ours | $0.514 \pm 0.016$
$0.0 \pm 0.0$
$0.0 \pm 0.0$
$0.0 \pm 0.0$
$0.485 \pm 0.016$
$0.0 \pm 0.0$ | $12.0 \pm 0.0$
$6.12 \pm 0.603$
$8.354 \pm 0.135$
$6.192 \pm 0.19$
$11.868 \pm 0.04$
$10 \pm 0$ | $5.19 \pm 0.158$
$3.25 \pm 0.313$
$4.505 \pm 0.067$
$3.61 \pm 0.105$
$4.975 \pm 0.157$
$5 \pm 0$ |
| C-Tiger

$(\hat{c} = 1.5)$ | 2 / 3 / 2 | CGCP
CGCP-CL
CPBVI
CPBVI-D
EXP-Gradient
Ours | $0.674 \pm 0.015$
$0.76 \pm 0.043$
$0.482 \pm 0.016$
$0.0 \pm 0.0$
$1.0 \pm 0.0$
$0.0 \pm 0.0$ | $-62.096 \pm 3.148$
$-72.424 \pm 5.283$
$-74.456 \pm 1.79$
$-75.414 \pm 1.617$
$-3.713 \pm 0.92$
$-75.075 \pm 1.511$ | $1.536 \pm 0.034$
$1.535 \pm 0.005$
$1.489 \pm 0.011$
$1.497 \pm 0.0$
$2.294 \pm 0.004$
$1.422 \pm 0.0$ |
| C-Tiger

$(\hat{c} = 3)$ | 2 / 3 / 2 | CGCP
CGCP-CL
CPBVI
CPBVI-D
EXP-Gradient
Ours | $0.753 \pm 0.014$
$0.140 \pm 0.035$
$0.153 \pm 0.011$
$0.0 \pm 0.0$
$1.0 \pm 0.0$
$0.0 \pm 0.0$ | $-1.690 \pm 0.647$
$-2.983 \pm 2.045$
$-11.11 \pm 1.05$
$-178 \pm 2.62$
$1.813 \pm 0.323$
$-5.75 \pm 0.522$ | $2.996 \pm 0.014$
$2.930 \pm 0.035$
$2.58 \pm 0.010$
$0.0 \pm 0.0$
$3.222 \pm 0.007$
$2.982 \pm 0.001$ |
| CRS(4,4)

$(\hat{c} = 1)$ | 201 / 8 / 3 | CGCP
CGCP-CL
CPBVI
CPBVI-D
EXP-Gradient
Ours | $0.512 \pm 0.024$
$0.78 \pm 0.004$
$0.0 \pm 0.0$
$0.0 \pm 0.0$
$0.295 \pm 0.014$
$0.0 \pm 0.0$ | $10.434 \pm 0.125$
$1.657 \pm 0.315$
$-0.4 \pm 0.316$
$-0.321 \pm 0.434$
$10.383 \pm 0.156$
$6.52 \pm 0.316$ | $0.512 \pm 0.016$
$0.724 \pm 0.040$
$0.522 \pm 0.016$
$3.082 \pm 0.005$
$0.918 \pm 0.058$
$0.523 \pm 0.016$ |
| CRS(5,7)

$(\hat{c} = 1)$ | 3201 / 12 / 3 | CGCP
CL-CGCP
CPBVI
CPBVI-D
EXP-Gradient
Ours | $0.412 \pm 0.022$
$0.18 \pm 0.009$
$0.0 \pm 0.0$
$0.0 \pm 0.0$
$0.30 \pm 0.014$
$0.0 \pm 0.0$ | $11.984 \pm 0.193$
$9.641 \pm 0.477$
$0.0 \pm 0.0$
$0.0 \pm 0.0$
$11.90 \pm 0.22$
$11.766 \pm 0.137$ | $1.00 \pm 0.038$
$0.991 \pm 0.034$
$0.0 \pm 0.0$
$0.0 \pm 0.0$
$1.31 \pm 0.06$
$0.950 \pm 0.0$ |
| CRS(7,8)

$(\hat{c} = 1)$ | 12545 / 13 / 3 | CGCP
CL-CGCP
CPBVI
CPBVI-D
EXP-Gradient
Ours | $0.357 \pm 0.015$
$0.20 \pm 0.13$
$0.0 \pm 0.0$
$0.0 \pm 0.0$
$0.322 \pm 0.015$
$0.0 \pm 0.0$ | $10.78 \pm 0.19$
$11.17 \pm 1.53$
$0.0 \pm 0.0$
$0.0 \pm 0.0$
$10.03 \pm 0.16$
$6.61 \pm 0.22$ | $0.945 \pm 0.04$
$0.931 \pm 0.078$
$0.0 \pm 0.0$
$0.0 \pm 0.0$
$1.154 \pm 0.054$
$0.960 \pm 0.003$ |
| CRS(11,11)

$(\hat{c} = 1)$ | 247809 / 16 / 3 | CGCP
CL-CGCP
CPBVI
CPBVI-D
EXP-Gradient
Ours | $0.0 \pm 0.0$
-
$0.0 \pm 0.0$
$0.0 \pm 0.0$
$0.0 \pm 0.0$
$0.0 \pm 0.0$ | $5.987 \pm 0.0$
-
$0.0 \pm 0.0$
$0.0 \pm 0.0$
$5.987 \pm 0.0$
$5.987 \pm 0.0$ | $0.0 \pm 0.0$
-
$0.0 \pm 0.0$
$0.0 \pm 0.0$
$0.0 \pm 0.0$
$0.0 \pm 0.0$ |
| Tunnels

$(\hat{c} = 1)$
P(correct obs) = 0.8 | 53 / 3 / 5 | CGCP
CL-CGCP
CPBVI
CPBVI-D
EXP-Gradient
Ours | $0.50 \pm 0.015$
$0.31 \pm 0.046$
$0.90 \pm 0.009$
$0.89 \pm 0.010$
$0.48 \pm 0.016$
$0.0 \pm 0.0$ | $1.612 \pm 0.011$
$1.22 \pm 0.058$
$1.921 \pm 0.0$
$1.921 \pm 0.0$
$1.35 \pm 0.02$
$1.028 \pm 0.018$ | $1.011 \pm 0.016$
$0.683 \pm 0.082$
$1.621 \pm 0.02$
$1.568 \pm 0.024$
$0.815 \pm 0.03$
$0.440 \pm 0.020$ |
| Tunnels

$(\hat{c} = 1)$

P(correct obs) = 0.95 | 53 / 3 / 5 | CGCP
CL-CGCP
C-PBVI
EXP-Gradient
CPBVI-D
Ours | $0.492 \pm 0.016$
$0.27 \pm 0.045$
$0.783 \pm 0.013$
$0.44 \pm 0.016$
$0.812 \pm 0.012$
$0.0 \pm 0.0$ | $1.679 \pm 0.008$
$1.17 \pm 0.061$
$1.921 \pm 0.0$
$1.42 \pm 0.02$
$1.921 \pm 0.0$
$1.010 \pm 0.017$ | $1.056 \pm 0.033$
$0.812 \pm 0.069$
$1.517 \pm 0.026$
$0.86 \pm 0.03$
$1.57 \pm 0.024$
$0.273 \pm 0.013$ |

Table 3: Comparison of our RC-POMDP algorithm to state-of-the-art offline C-POMDP algorithms. We report the mean and 1 standard error of the mean for each metric. A memory-out is indicated by a $-$. Note that for CRS(11,11), due to the $300s$ time limit, CGCP, Exp-Gradient and ours all compute a policy that goes directly to the exit area without interacting with any rocks, and achieve the same reward and 0 cost.