# OpenReview forum: "Recursively-Constrained Partially Observable Markov Decision Processes"
_auai.org/UAI/2024/Conference — UAI 2024 oral_

### Official Review · Reviewer_abPA · 2024-03-21

**Q2-1 Originality-Novelty:** 3
**Q2-2 Correctness-Technical Quality:** 3
**Q2-5 Clarity Of Writing:** 3

**Q10 Ethical Concerns:**

No.

**Q1 Summary And Contributions:**

The paper showcases the unintuitive nature of Constrained POMDPs (C-POMDPs) as they don't obey the Bellman's principle of optimality, and optimal policies in C-POMDPs can have potentially unwanted behaviour. The paper therefore introduces Recursively-Constrained POMDPs (RC-POMDPs) to counter these drawbacks. The main idea is to impose recursive constraints on beliefs for future time steps. The paper presents some useful theoretical properties of RC-POMDPs. The authors present an algorithm to solve RC-POMDPs w.r.t. (discounted) infinite horizon total reward objectives and evaluate it on several benchmarks.

**Q2-3 Extent To Which Claims Are Supported By Evidence:**

3: Good: the main claims are supported by convincing evidence (in the form of adequate experimental evaluation, proofs, (pseudo-)code, references, assumptions).

**Q2-4 Reproducibility:**

3: Good: key resources (e.g. proofs, code, data) are available and key details (e.g. proofs, experimental setup) are sufficiently well-described for competent researchers to confidently reproduce the main results.

**Q3 Main Strengths:**

I really like the motivation for the paper, and the topic of the paper seems relevant to current research efforts in this area. I think the motivation is well presented with a good and straightforward example. The paper presents a novel problem formulation building a good foundation for possible future research. Technically the paper is strong. I appreciate the presentation of the algorithms as well as the theoretical results and all of the corresponding proofs, and I have no reason to doubt their correctness. I like the presentation of the differences between the policies for C-POMDP and RC-POMDP formulations in Figure 2.

**Q4 Main Weakness:**

Even though I understand RC-POMDPs are a new problem presented in this paper, I find the presented problems considered in the experimental evaluation a bit trivial. Could the authors add claims about the scalability potential of their approach? Maybe with the inclusion of larger benchmarks, it might also be interesting to consider other baseline algorithms (Wray and Czuprynski ICRA'22 or Poupart et al.~AAAI'15). Other than that, I have just minor points of contention.

**Q5 Detailed Comments To The Authors:**

I think in Example 1, the description of rewards is missing. Without this, it seems strange why the rover cannot just take tunnel B since it has probability 1 of being without rocks.

I think the strong point of the paper is the main motivation that C-POMDPs are not good for producing intuitive policies. However, the experimental section doesn't fully explore this aspect other than Figure 2 and the violation rate metric (which I don't like that much). Maybe (at least from my point of view) this is a more interesting part of the evaluation, and it could be interesting to present this in small case studies. It could also be interesting to showcase which models from the traditional C-POMDP benchmarks showcase strong cases of stochastic self-destruction.

I would advise adding a table that summarises the considered benchmarks i.e. number of states/actions/observations (at least in the appendix if there's no room).

I would also like the authors to address the potential future work in conclusion. I really like the new concepts presented, and it seems natural for me to propose how this can be used/expanded on or what are your plans.

**Q9 Complying With Reviewing Instructions:**

Yes

---

> ### Author Rebuttal · Authors · 2024-04-07
>
> We thank the reviewer for the insightful comments and questions.
>
> **Q4.1: Scalability.**
>
> That is a good point. You are correct that the primary focus of the work is the discussion on C-POMDPs and the RC-POMDP formulation; the algorithm is the secondary focus of the paper. That is why we decided to use smaller problems for evaluations which make illustrations and comparisons easier.
>
> Currently, the algorithm scales to moderately sized problems. The largest problem we considered in the paper is CRS(5,7), which has a state, action, and observation space of (3201, 12, 3). The largest problem we found that we could scale to is CRS(11,11), which has a state, action, observation space of (247809, 16, 3), which our algorithm obtains a reward-value of 5.99 in 300s. Our algorithm finds an admissible policy but does not converge given the short time limit. Since we also wanted to compare with CGCP-CL which takes the full time limit for each action call, increasing the time limit was prohibitively time intensive for the main purpose of comparing the behaviors of RC-POMDPs to C-POMDPs and seeing the prevalence of the stochastic self-destruction behavior of C-POMDP policies. We will add results for CRS(11,11) to the final version of the paper.
>
> We emphasize that the problems our algorithm can currently solve are large enough to be useful in real-life safety-critical applications, such as route decisions for navigation-based problems and high-level decisions for robotic search and rescue.
>
> **Q4.2: Other baseline algorithms.**
>
> Thanks for the suggestion. For a few reasons, we believed our chosen comparison algorithms were adequate for the purpose of our initial evaluation. CGCP finds optimal C-POMDP policies, and CGCP-CL evaluates the replanning performance of CGCP. These algorithms support our theoretical discussion. CPBVI and CPBVI-D are the closest to our algorithm with a point-based value iteration technique. These algorithms illustrate the performance of our algorithm compared to similar methods.
>
> We considered comparing to CALP (Poupart et al. 2015), but since CGCP is shown to be more performant with better scalability, we focused on CGCP. The projected gradient ascent (PGA) technique of Wray and Czuprynski 2022 is very interesting. Since the algorithm solves C-POMDP, we would expect that the policies still exhibit stochastic self-destruction behaviors.
>
> That said, the PGA approach is promising, and it may be possible to design an algorithm for RC-POMDPs using PGA. We will add a discussion on Wray and Czuprynski 2022, and other techniques that compute controllers directly in our related work section. For future work, where the focus is the algorithm, we will compare to a wider class of algorithms and larger benchmark problems.
>
> **Q5.1: Description of rewards**
>
> That is a very good suggestion. We will add a description of reward that incentivizes traversing tunnel A.
>
> **Q5.2: More case studies on stochastic self-destruction.**
>
> Thanks for the insightful comment. Although we agree that the violation rate metric is not a perfect one, it is still informative in that it shows the percentage of trials that exhibit the stochastic self-destruction behavior.
>
> From our current benchmarks, all of the problems showcase existence of stochastic self-destruction behavior. The difference between the violation rate of CGCP and CGCP-CL (CGCP with closed-loop replanning) is higher for some models, which might signal that these problems showcase stronger effects of violation of the optimal substructure property, and thus stochastic self destruction.
>
> Per your suggestion, we plan to study which models showcase strong cases of stochastic self-destruction, and develop more metrics that signal stochastic self-destruction.
>
> **Q5.3: Summary of benchmarks**
>
> We will add a summary of the considered benchmarks with the sizes of their (state, actions, observations): C-Tiger: (2, 3, 2), CE: (5, 2, 2), Tunnels: (53, 3, 5), CRS(4,4): (201, 8, 3), CRS(5,7): (3201, 12, 3), CRS(11,11): (247809, 16, 3).
>
> **Q5.4: Future Work**
>
> Thanks for the suggestion. We will add a section on future work as we believe RC-POMDP opens up many avenues for research. Here, we provide a (non-exhaustive) summary.
>
> 1. Analyzing classes (or conditions) of RC-POMDPs that are approximable.
>
> 2. RC-POMDPs provide arguably more desirable policies than C-POMDPs, but the cost constraints remain on expectation. For some applications, probabilistic or risk measure constraints may be more desirable than expectation constraints. These formulations also benefit from the recursive constraints that we propose for RC-POMDPs.
>
> 3. Scalability: Our algorithm can benefit from better policy search heuristics and more efficient policy representations. We also plan to explore other methods, such as finite state controllers (e.g. PGA), occupation measures, and other approximation methods (e.g. online tree search (Lee et al. 2018, Jamgochian et al. 2023, etc)).

---

### Official Review · Reviewer_tSMM · 2024-03-23

**Q2-1 Originality-Novelty:** 3
**Q2-2 Correctness-Technical Quality:** 4
**Q2-5 Clarity Of Writing:** 4

**Q1 Summary And Contributions:**

Constrained MDPs have been widely studied and their extension to POMDPs faces some difficulties that this paper addresses. First, it's shown that C-POMDPs violate the optimal substructure property. In this paper the authors introduce RC-POMDP (Recursively-Constrained POMDPs) to overcome this difficulty and to restore the optimal substructure property. The authors identify the issue and characterize it as a stochastic self-destruction. The source of such behaviors come from the history-dependent POMDP policies and thus some basis of MDP solving algorithms are not valid. The authors identify well the source of the problem by noticing that C-POMDPs are not well formalized that  could lead to undesired problem. To this end, the authors introduce RC-POMDP where histories are considered in cost computation and propose a model where states are augmented by history to consider both the value and the cost of histories for optimisation. Furthermore, they transform this augmented model into an augmented belief MDP that classical algorithms can solve.

The paper is well written, the objectives are clearly stated and the solution are well motivated.

**Q2-3 Extent To Which Claims Are Supported By Evidence:**

3: Good: the main claims are supported by convincing evidence (in the form of adequate experimental evaluation, proofs, (pseudo-)code, references, assumptions).

**Q2-4 Reproducibility:**

3: Good: key resources (e.g. proofs, code, data) are available and key details (e.g. proofs, experimental setup) are sufficiently well-described for competent researchers to confidently reproduce the main results.

**Q3 Main Strengths:**

The paper states well the limites of C-POMDP and explains that the source of this limit comes from the formalisation of the C-POMDP that can exhibit a stochastic self-destructing. They propose a new formalisation named RC-POMDP to reconsider histories in the cost evaluation and propose a model where value function and the cost function are evaluated according to the histories. Then they transform the model to belief MDP to ease the solving while considering the cost of history. Theoretic analysis are provided.

**Q4 Main Weakness:**

Discussing other alternative Multi-Objective POMDPs to solve C-POMDP.

**Q5 Detailed Comments To The Authors:**

The Formalisation is clean and well structured. The proposed solution is interesting based on some existing ideas as the use of belief MDPs, augmenting the belief state with the history, ... I have two questions  on the proposed work :
- Why not considering a Multi-objective POMDPs by considering a vector of value  (R, C) by considering as soon as C is greater than a threshold, C becomes infinite ? At least a discussion is needed.
- In experimental evaluation, I would like to see an experiment with the threshold is very large as an unconstrained POMPDs to compare the obtained policies with an optimal policy of a POMDP.

**Q9 Complying With Reviewing Instructions:**

Yes

---

> ### Author Rebuttal · Authors · 2024-04-07
>
> We thank the reviewer for the helpful comments and questions.
>
> **Q4/Q5.1: Why not consider a Multi-objective POMDP by considering a vector of value (R, C) by considering as soon as C is greater than a threshold, C becomes infinite?**
>
> Thanks for this question. It helps us to explain our method from the Multi-objective POMDP (MO-POMDP) perspective. In essence, our algorithm performs exactly the suggested approach. At each node of the tree, we keep track of upper and lower bounds on R and C. As soon as the lower bound on C is greater than a threshold, that part of the search space is pruned from consideration, and the admissibility horizon set to 0. This is equivalent to saying C becomes infinite. Performing your suggestion would be functionally equivalent to our current algorithm. The only difference is that we also keep track of the admissibility horizon, which is useful for explainability purposes for the case when no admissible policy exists.
>
> It is also worth noting that the suggested MO-POMDP perspective explains our algorithmic approach well, but the problem it solves is still the RC-POMDP problem, and not the C-POMDP problem. We will include a discussion on the MO-POMDP perspective in the final version of the paper.
>
> **Q5.2: Experiment with very large cost threshold (so as to be equivalent to an unconstrained POMDP).**
>
> Thank you for the suggestion! This is an interesting question that touches on how generalizable and efficient our RC-POMDP algorithm and other C-POMDP algorithms are for problems that are less constraining. Since all policies are admissible in such a case (and so our algorithm does not have issues with conservatism), our algorithm is guaranteed to asymptotically converge to the optimal solution, but we would expect it to converge at a slower rate than a "fast" unconstrained POMDP algorithm.
>
> Since our algorithm needs to keep track of admissible cost values, we mainly use a policy tree representation. This representation is less efficient than the $\alpha$-vector policy representation used in SARSOP and other "fast" offline unconstrained POMDP algorithms, which allow value improvements at a belief state to directly improve values at other belief states. Therefore, we would expect that the algorithm is less efficient in converging to near-optimal solutions than a more specialized unconstrained POMDP algorithm and a C-POMDP algorithm that utilizes these unconstrained POMDP algorithms.
>
> To answer this question, we have conducted some additional preliminary experiments. Due to time and computation constraints, we focused on our algorithm (RC-POMDP), CGCP (optimal C-POMDP), and SARSOP (Kurniawati et al. 2008) (unconstrained POMDP algorithm). We will include a more comprehensive empirical analysis in the final version of the paper.
>
> |                 | Ours (RC-POMDP)           | CGCP (C-POMDP)          | Unconstrained POMDP (SARSOP) |
> |-----------------|----------------|----------------|------------------------------|
> |                 | (Reward, Cost) | (Reward, Cost) | Reward                       |
> | C-Tiger (300s time limit)          | (-1.4, 3.2)    | (1.90, 3.2)     | 1.93                         |
> | CE (300s time limit) | (12.0, 4.5)    | (12.0, 5.0)    | 12.0                         |
> | Tunnels (300s time limit)         | (1.92, 1.6)    | (1.92, 1.6)   | 1.92                         |
> | CRS(4,4) (300s time limit) | (16.9, 2.2)    | (16.9, 2.4)    | 16.9                         |
> | CRS(5,7) (300s time limit)| (14.9, 2.1)    | (14.8, 3.6)    | 23.9 |
> | CRS(5,7) (1000s time limit)| (15.3, 2.2)    | (24.0, 4.5)    | 24.0 |
>
> As seen in the table, our algorithm performs similar to CGCP and the unconstrained POMDP algorithm SARSOP for most smaller problems. The C-Tiger problem benefits greatly from the $\alpha$-vector representation, since the optimal policy repeatedly cycles among a small set of belief states (which our algorithm considers different augmented belief-admissible cost states). For slightly larger problems (CRS(5,7)), the efficient $\alpha$-vector representation and other heuristics of SARSOP (which CGCP takes advantage of, since it repeatedly calls SARSOP) enables much faster convergence than the policy tree-based method of our approach. Nonetheless, as time is increased, our algorithm does slowly but surely improve values.
>
> An interesting future direction would be to look into how an RC-POMDP algorithm can take advantage of the relative relevance of cost to reward in different parts of the search space. Note that algorithms like CGCP and CC-POMCP (Lee et al. 2018) are implicitly guided by the relative relevance of cost through the dual formulation (R - $\lambda$C). For an effectively unconstrained problem, these C-POMDP algorithms can converge relatively quickly by finding the optimal $\lambda^* = 0$, which reduces to the unconstrained POMDP problem.

---

### Official Review · Reviewer_JAak · 2024-03-25

**Q2-1 Originality-Novelty:** 3
**Q2-2 Correctness-Technical Quality:** 3
**Q2-5 Clarity Of Writing:** 3

**Q1 Summary And Contributions:**

This paper proposes an alternative for the standard constrained POMDP framework. This framework essentially imposes (infinitely) more stringent constraints which may be suitable for some safety-critical applications. The paper provides a solution methodology for the new framework and a fairly rigor analysis of this solution.

**Q2-3 Extent To Which Claims Are Supported By Evidence:**

2: Fair: the main claims are somewhat supported by evidence (but the experimental evaluation may be weak, or does not match entirely with the claims, important baselines may be missing, proofs contain important ideas but lack rigor, algorithmic details are only discussed superficially, references are imprecise, assumptions are not sufficiently motivated or explicated, etc.).

**Q2-4 Reproducibility:**

3: Good: key resources (e.g. proofs, code, data) are available and key details (e.g. proofs, experimental setup) are sufficiently well-described for competent researchers to confidently reproduce the main results.

**Q3 Main Strengths:**

I think the paper has some interesting ideas in terms of how to formulate constraints in safety-critical applications. The framework and solution are backed up with rigorous analysis. I could not find any major errors in their proofs.

**Q4 Main Weakness:**

1. The constraints could be too stringent.
2. For the motivation provided, the proposed framework may not be necessary. This raises a question of complicating the problem without necessity.
3. Some important existing approaches have not been included in the comparison study.

**Q5 Detailed Comments To The Authors:**

When I read Example 1 (cave navigation), it felt like the problem could be easily solved using other methods instead of using the method proposed in the paper. If avoiding rocky terrain is so important, why not simply increase the cost to say 100*(1/(1-\gamma))? Further, if a no-regret learning method such as the one in reference (a) below was used, I think it would automatically find a solution that avoids rocky terrain. This is because this approach solves a bunch of unconstrained optimization problems of the form reward-lambda*cost. So there is a natural bias for solutions with lower cost. Furthermore, if an admissible policy exists, then the no-regret learning approach also would likely come up with a deterministic solution. This approach has no online replanning issues because it uses unconstrained solutions in which online replanning is straightforward.

What would really convince me if the paper has an example for which no existing solution works.

a. Kalagarla, K. C., Dhruva, K., Shen, D., Jain, R., Nayyar, A., & Nuzzo, P. (2022, August). Optimal control of partially observable Markov decision processes with finite linear temporal logic constraints. In Uncertainty in Artificial Intelligence (pp. 949-958). PMLR.

**Q9 Complying With Reviewing Instructions:**

Yes

---

> ### Author Rebuttal · Authors · 2024-04-07
>
> We thank the reviewer for the helpful comments and questions.
>
> **Q4.1: Constraints too stringent.**
>
> RC-POMDPs are indeed more stringent than C-POMDPs, but we disagree that they are "infinitely more stringent" than C-POMDPs (as mentioned in the reviewer's summary in Q1). We stress that the imposed constraints are designed to be stringent enough to circumvent the unintuitive behavior of C-POMDP policies, but not as stringent as a worst-case constraint. In fact, as discussed in Remark 2 in Section 3, RC-POMDP falls in between the two extreme cases. C-POMDPs bound the expected total cost of state trajectories, enabling belief trajectories with low expected cost to compensate for high expected cost ones. Conversely, a worst-case constraint formulation, which never allows any violations during execution, may be overly conservative. RC-POMDPs strike a balance between the two; it bounds the expected total cost for all belief trajectories, only allowing cost violations during execution due to state uncertainty. This is illustrated by our experiments, which generally show that, while RC-POMDPs policies avoid pathological behaviors, their rewards remain competitive with the rewards of optimal C-POMDP policies.
>
> **Q4.2/Q5: Why not simply increase cost?**
>
> Thank you for the question. Increasing cost is a practical method for many situations. However, there are two main limitations of just increasing the cost:
>
> 1. Increasing cost doesn't prevent the agent from traversing tunnel A. For example, suppose one policy (traverse tunnel B) incurs 0 cost and reward, while another policy (traverse tunnel A) has high reward but incurs constraint-violating cost. Then, a C-POMDP optimal policy allocates as much maximum probability to the high cost policy as possible without violating the constraint. The cost for tunnel A can be made arbitrarily large, and the resulting mixed policy reduces the probability of traversing A, but it remains nonzero. In Example 1, if the cost of tunnel A is increased to a large value $c$, the expected cost is $0.8c$. With cost threshold $0 < \hat{c} < \infty$, the optimal C-POMDP policy traverses A with probability $\frac{\hat{c}}{0.8c}$ and traverses B with probability $1 - \frac{\hat{c}}{0.8c}$. The unintuitive behavior of C-POMDPs is not addressed by simply increasing cost.
>
> 2. A POMDP is inherently characterized by state uncertainty. If cost in a state is too high (so as to behave like a worst-case constraint), the C-POMDP problem may be infeasible as long as some probability mass is in that state, as all policies violate the constraints. Thus, increasing cost can become overly conservative.
>
> In many partially observable (even safety-critical) problems, there may be undesirable state-actions best modeled with expected costs thresholds, rather than worst-case constraints. An example of this is a sample exploration mission in which there is some risk of getting stuck when traversing some partially observable terrain, but traversal may be necessary for mission completion. This motivates RC-POMDPs, which strike a balance between C-POMDPs and worst-case constraints.
>
> **Q4.3/Q5: Comparison with no-regret learning approach.**
>
> Thank you for the interesting and important question. The mentioned work (Kalagarla et al. 2022) performs a similar primal-dual approach, with a different dual update procedure, as the CGCP algorithm (Walraven and Spaan 2008) discussed and compared to in our paper. CGCP solves a sequence of unconstrained POMDP optimizations in the form of $R - \lambda C$, and finds optimal (mixed) policies for C-POMDPs. In contrast, Kalagarla et al. 2022 finds a deterministic policy, but may be suboptimal for C-POMDPs as optimal policies for C-POMDPs are generally stochastic (Kim et al. 2011). In a sense, Kalagarla et al. 2022 restricts the search to deterministic policies.
>
> In our counter-example (CE) in Figure 1, which is an abstracted variation of Example 1, the optimal C-POMDP policy is deterministic. The algorithm of Kalagarla et al. 2022 would compute exactly that optimal policy. However, as discussed in Section 2.1, that policy exhibits stochastic self-destruction.
>
> Empirically, from Table 1 in Section 6, CGCP finds the optimal C-POMDP deterministic policy. There exists an admissible policy for CE, found by our algorithm, but CGCP always decides to traverse tunnel A. The result for CGCP-CL (CGCP with replanning) in CE shows that there are indeed replanning inconsistencies stemming from the violation of the optimal substructure property, even for problems which have deterministic optimal policies (or when restricted to deterministic policies). Therefore, in CE and in general, Kalagarla et al. 2022 does not find admissible policies and would have replanning issues.
>
> In summary, the algorithm of Kalagarla et al. 2022 still suffers from the shortcomings of the C-POMDP formulation. We will discuss this, and compare to it, in the final version of the paper.

---

### Official Review · Reviewer_PKtD · 2024-03-26

**Q2-1 Originality-Novelty:** 3
**Q2-2 Correctness-Technical Quality:** 3
**Q2-5 Clarity Of Writing:** 3

**Q1 Summary And Contributions:**

This paper studies constrained partially observable Markov decision processes (C-POMDPs) model. The authors first show that C-POMDPs violate the optimal substructure property over successive decision steps and thus may exhibit behaviors that are undesirable for some applications. Then, online re-planning in C-POMDPs is often ineffective due to the inconsistency resulting from this violation. To address these drawbacks, they introduce the Recursively-Constrained POMDP (RC-POMDP), which imposes additional history-dependent cost constraints on the C-POMDP. They show that RC-POMDPs always have deterministic optimal policies and that optimal policies obey Bellman’s principle of optimality. They present a point-based dynamic programming algorithm for RC-POMDPs. Evaluations on benchmark problems demonstrate the efficacy of the proposed algorithm.

**Q2-3 Extent To Which Claims Are Supported By Evidence:**

3: Good: the main claims are supported by convincing evidence (in the form of adequate experimental evaluation, proofs, (pseudo-)code, references, assumptions).

**Q2-4 Reproducibility:**

3: Good: key resources (e.g. proofs, code, data) are available and key details (e.g. proofs, experimental setup) are sufficiently well-described for competent researchers to confidently reproduce the main results.

**Q3 Main Strengths:**

This paper is clearly written and well-structured. It provides a detailed analysis that the constrained POMDPs do not have the optimal substructure property over successive decision steps and the consequences, which provides a good insight for the constrained POMDP problem. Moreover, the authors tackle the challenge by defining a novel POMDP problem with constraints named RC-POMDP and further prove a good property of RC-POMDP. Finally, the authors provide a dynamic programming algorithm that can solve the proposed new POMDP problem and conduct the empirical evaluation. Overall, the contribution of this work is significant. The problem of POMDPs with constraints is well-explored in this paper.

**Q4 Main Weakness:**

I did not identify any major weaknesses in the paper. However, my evaluation is based on my limited knowledge of this work's topic.

**Q5 Detailed Comments To The Authors:**

N/A

**Q9 Complying With Reviewing Instructions:**

Yes

---

> ### Author Rebuttal · Authors · 2024-04-07
>
> We thank the reviewer for your encouraging review. We are glad that the reviewer finds the contribution of our work significant.

---

### Official Review · Reviewer_GJpu · 2024-03-27

**Q2-1 Originality-Novelty:** 3
**Q2-2 Correctness-Technical Quality:** 3
**Q2-5 Clarity Of Writing:** 3

**Q1 Summary And Contributions:**

This paper addresses the problem of constrained POMDPs. Constrained MDPs (and POMDPs) do not satisfy Bellman's principle of optimality, and hence have to solved via convex analytic methods via introduction of an occupation measure. This makes the corresponding LP computationally prohibitive or impossible to solve when the state is very large, or infinite. In the case of POMDPs, the belief state space is always uncountable, so there is a special difficulty. Another problem is that C-POMDPs involve optimization over occupations measure of belief states: and the only way to get an approximate solution would be a function approximation of the occupation measure. This paper attempts to skirt these issues by making a stricter requirement on constraints being satisfied along all trajectories, at all times. These are called RC-POMDPs. They show that RC-POMDPs have at least one deterministic optimal policy, satisfy Bellman's principle of optimality (in some sense) and the Bellman operator has a unique fixed point. They introduce a  dynamic programming algorithm, and show via empirical results that their algorithm achieves better total reward while ensuring "recursive constraint" always satisfied.

**Q2-3 Extent To Which Claims Are Supported By Evidence:**

3: Good: the main claims are supported by convincing evidence (in the form of adequate experimental evaluation, proofs, (pseudo-)code, references, assumptions).

**Q2-4 Reproducibility:**

3: Good: key resources (e.g. proofs, code, data) are available and key details (e.g. proofs, experimental setup) are sufficiently well-described for competent researchers to confidently reproduce the main results.

**Q3 Main Strengths:**

+ The paper presents an interesting idea to solve C-POMDPs approximately by actually tightening the requirement on constraint satisfaction.

+ The algorithm is well supported by mathematical theory.

+ experimental results are convincing of the claims.

**Q4 Main Weakness:**

- A key issue introduced is that instead of the policy being a function of a sufficient statistic like the belief state, it is now a function of the entire history. This when the horizon is long enough, or the state-action space large enough is very undesirable. This is the main reason why we introduce policies as functions of belief states instead of histories in POMDPs.

- Another key issue introduced is that the state space has to be augmented with current "cumulative cost". Again, this is very undesirable. This can be inevitable in control problems with risk measures, but is not necessarily unavoidable in C-POMDPs, by which I mean that there are other approximation avenues available.

- In some sense Theorem 2 and 3 are not surprising once the state space is augmented, and the Bellman operator is defined in that particular way.

- Experimental results are not surprising since the other algorithms have been designed with different objectives in mind.

**Q5 Detailed Comments To The Authors:**

1. There is typo in Theorem 3 (asymptotic claim).

2. I have not checked the algorithms you are comparing against. Does any of them correspond to function-approximating the occupation measure of the belief state? Have you tried it? Does it work? How does it compare?

**Q9 Complying With Reviewing Instructions:**

Yes

---

> ### Author Rebuttal · Authors · 2024-04-07
>
> We thank the reviewer for the thoughtful comments and questions.
>
> **Q4.1: Policy as a function of history.**
>
> Thanks for the comment, which gives us an opportunity to clarify. For an RC-POMDP, we do not need to explicitly store the full history. Since the evolution of the augmented belief-admissible cost state $\bar{b}_t$ is Markovian, and due to the satisfaction of BPO in RC-POMDPs, only a belief and admissible cost are needed to execute the policy. In other words, $\bar{b}_t$ is a sufficient statistic of the history for RC-POMDP. We use $h_t$ in the equations for RC-POMDP problem formulation and other theoretical discussions for clarity and notation consistency.
>
> It should also be noted that, like all HSVI-based algorithms, our algorithm uses a tree that implicitly contains the history in the path from the root node to another node, but the policy produced by the algorithm is still sound outside this tree. In the final version, we will clarify that history is not needed for policy execution.
>
> **Q4.2: State space has to be augmented with current "cumulative cost".**
>
> We would like to clarify that we do not augment the state space with cumulative cost. Instead, we directly augment the belief space with cumulative cost. A belief state is a simplex over the states, and therefore the size of a belief state is $n$ for state spaces of $n$ cardinality. Since the number of costs is generally significantly lower than the state space cardinality, this belief space augmentation incurs insignificant performance costs. Therefore, the search space of the problem is not increased significantly.
>
> **Q5.2: Do any of them (compared algorithms) correspond to function-approximating the occupation measure of the belief state? Have you tried it? Does it work? How does it compare?**
>
> Thanks for this comment, which made us think carefully about approaches based on occupation measure. An optimal policy of a C-POMDP can be obtained by solving an LP using a discounted occupancy measure (Altman 1999, Lee et al. Neurips'18). However, solving this LP exactly is intractable since the belief space may have infinite cardinality. Instead, existing methods solve the dual LP by computing a optimal solution $\lambda^*$. CC-POMCP (Lee et al. 2018) uses Monte-Carlo tree search with a subgradient method to approximate $\lambda^*$, while CGCP (Walraven and Spaan 2018) uses a column generation method to solve the LP defined over the entire policy space.
>
> CC-POMCP performs well, but only has asymptotic guarantees on constraint satisfaction and convergence. CGCP enjoys convergence to optimality, and computes sound solutions. Since our work focuses on the problem formulations of C-POMDP and RC-POMDP with guarantees on constraint satisfaction, we compare to CGCP in our evaluation. In our evaluation, CGCP exhibits the drawbacks of C-POMDP policies, validating our theory and justification for RC-POMDPs. This is because the unintuitive behavior stems from the C-POMDP formulation itself, rather than the method for solving it.
>
> For RC-POMDP, it is not clear to us how to formulate a method that uses occupancy measure with the recursive cost constraints. Instead, we develop a tree-based policy search algorithm that finds admissible policies, in order to validate our theory and proposed RC-POMDP formulation.
>
> **Q4.3: In some sense Theorem 2 and 3 are not surprising once the state space is augmented, and the Bellman operator is defined in that particular way.**
>
> Yes, the reviewer is correct, and it is in fact by design. One of the main focuses of our work is to provide a problem formulation in which the Bellman's principle of optimality can be satisfied, which allows consistency of planning over successive decision steps to mitigate the discussed drawbacks of C-POMDP policies and enables dynamic programming techniques.
>
> **Q4.4: Experimental results are not surprising since the other algorithms have been designed with different objectives in mind.**
>
> Yes, the reviewer is again correct in that the results follow and validate our theoretical analysis and highlight advantages of the proposed problem/approach. We stress that the purpose of our experiments is to evaluate the behavior of C-POMDP policies on common POMDP problems with constraints. In particular, the C-RockSample problem is a C-POMDP benchmark problem from (Lee et al. 2018). These results serve as examples of POMDPs in which C-POMDP solutions exhibit the undesirable (``stochastically self-destructive") behavior, thereby validating our theoretical discussion and justification for proposing RC-POMDPs.
>
> **Q5.1: There is typo in Theorem 3 (asymptotic claim).**
>
> Thank you for catching this. We will update the asymptotic result of $V_C^{\pi^n}$ to $V_C^{\pi^\infty}$. Note that although $V_R^{\pi^*}$ is a unique fixed point, $V_C^{\pi^\infty}$ does not converge to a fixed point $V_C^{\pi^*}$, since an optimal reward-value function may have multiple cost-value functions that satisfy the cost constraints.

---

### Meta-Review · Area_Chair_y3za · 2024-04-17

This paper addresses a problem in constrained POMDPs in that standard solution concepts can violate substructure optimality and lead to unintuitive behaviors. A new recursively constrained formulation addresses the issue in compelling fashion and is complemented with some theoretical analysis (proving nice properties) and some reasonable empirical evaluation. All of the reviewers were positive on the paper to varying degrees. That said, the reviewers posed some questions and made several suggestions that should be used to improve the paper. The authors are encouraged to make clarifications/revisions (e.g., as in the rebuttal) to prevent misreadings or misunderstandings by the reader.

All in all, this would make a nice contribution to UAI.